# Neurobiological and Existential Profiles in Posttraumatic Stress Disorder: The Role of Serotonin, Cortisol, Noradrenaline, and IL-12 Across Chronicity and Age

**DOI:** 10.3390/ijms26199636

**Published:** 2025-10-02

**Authors:** Barbara Paraniak-Gieszczyk, Ewa Alicja Ogłodek

**Affiliations:** Collegium Medicum, Jan Dlugosz University in Częstochowa, Waszyngtona 4/8 Street, 42-200 Czestochowa, Poland; eoglodek@gmail.com

**Keywords:** biomarkers, cortisol, death acceptance, existential attitudes, IL-12, life control, life purpose, noradrenaline, posttraumatic stress disorder, serotonin

## Abstract

Posttraumatic Stress Disorder (PTSD) is characterized by disruptions in central nervous system functioning and existential crises, yet the mechanistic links between neurobiological processes and dimensions of life meaning and identity remain underexplored. The aim of this study was to examine the relationships between stress biomarkers (serotonin, cortisol, noradrenaline, and interleukin-12 [IL-12]) and existential attitudes (measured using the Life Attitude Profile (Revised) [LAP-R]) in mining rescuers, considering PTSD duration and participant age. This cross-sectional study included 92 men aged 18–50 years, divided into three groups: no PTSD (n = 28), PTSD ≤ 5 years (n = 33), and PTSD > 5 years (n = 31). Serum levels of four biomarkers and LAP-R scores across eight domains were evaluated. Statistical analyses employed nonparametric tests, including the Kruskal–Wallis test for overall group differences (with Wilcoxon r effect sizes for pairwise comparisons, Mann–Whitney U tests for post hoc pairwise comparisons, and Spearman’s rank correlations for biomarker–LAP-R associations. Age effects were assessed in two strata: 18–35 years and 36–50 years. Kruskal–Wallis tests revealed significant group differences (*p* < 0.001) for all biomarkers and most LAP-R domains, with very large effect sizes (r > 0.7) in pairwise comparisons for serotonin (control median: 225.2 ng/mL vs. PTSD ≤ 5y: 109.9 ng/mL, r = 0.86; vs. PTSD > 5y: 148.0 ng/mL, r = 0.86), IL-12 (control: ~8.0 pg/mL vs. PTSD ≤ 5y: 62.4 pg/mL, r = 0.86; vs. PTSD > 5y: ~21.0 pg/mL, r = 0.69), and LAP-R scales such as Life Purpose (control: 54.0 vs. PTSD ≤ 5y: 39.0, r = 0.78; vs. PTSD > 5y: 20.0, r = 0.86) and Coherence (control: 53.0 vs. PTSD ≤ 5y: 34.0, r = 0.85; vs. PTSD > 5y: 23.0, r = 0.86). The PTSD ≤ 5y group exhibited decreased serotonin, cortisol (median: 9.8 µg/dL), and noradrenaline (271.7 pg/mL) with elevated IL-12 (all *p* < 0.001 vs. control), alongside reduced LAP-R scores. The PTSD > 5y group showed elevated cortisol (median: ~50.0 µg/dL, *p* < 0.001 vs. control, r = 0.86) and normalized IL-12 but persistent LAP-R deficits. Older participants (36–50 years) in the PTSD ≤ 5y group displayed improved existential attitudes (e.g., Life Purpose: 47.0 vs. 27.5 in 18–35 years, *p* < 0.001), whereas in PTSD > 5y, age exacerbated biological stress (cortisol: 57.6 µg/dL vs. 36.1 µg/dL, *p* = 0.003). Spearman correlations revealed stage-specific patterns, such as negative associations between cortisol and Death Acceptance in PTSD > 5y (ρ = −0.49, *p* = 0.005). PTSD alters biomarker levels and their associations with existential dimensions, with duration and age modulating patient profiles. These findings underscore the necessity for integrated therapies addressing both biological and existential facets of PTSD.

## 1. Introduction

Post-traumatic stress disorder (PTSD) is a serious mental disorder that develops as a result of experiencing traumatic events, both in personal and professional life, particularly in professions exposed to frequent contact with suffering, life-threatening situations, or extreme conditions. PTSD is characterized by persistent intrusive symptoms (recurring memories, flashbacks, nightmares), avoidance of stimuli reminiscent of the trauma, excessive psychophysiological arousal, and chronic difficulties in regulating emotions. According to the Diagnostic and Statistical Manual of Mental Disorders, Fifth Edition (DSM-5), diagnosis requires the presence of symptoms from four domains: intrusion, avoidance, negative cognitive-emotional changes, and hyperarousal [1,2,3]. In recent decades, significant progress has been made in understanding PTSD as a complex disorder with both psychological and biological foundations. Contemporary neurobiological models indicate dysfunction of the Hypothalamic–Pituitary–Adrenal (HPA) axis, persistent activation of the sympathetic nervous system, and dysregulated immune responses. Numerous studies have confirmed abnormalities in neurotransmitters, stress hormones, and inflammatory mediators, which may sustain PTSD symptoms and contribute to both mental and somatic complications [4,5,6].

This study focuses on four key biomarkers: serotonin, cortisol, noradrenaline, and interleukin 12 (IL-12), representing different yet interconnected neurobiological systems.

5-Hydroxytryptamine-Serotonin (5-HT) plays an essential role in regulating mood, sleep, impulsivity, aggression, and cognitive processes. In the context of PTSD, special significance is attributed to its influence on emotional memory, fear extinction, and adaptive behaviors [7,8,9]. Studies have shown reduced 5-HT levels in individuals with PTSD within limbic structures and the prefrontal cortex [10]. Animal models suggest that deficits in the serotonergic system—particularly in the raphe nucleus–hippocampus–prefrontal cortex axis—increase anxiety and impair contextual memory [11,12]. It is believed that a shift in receptor balance (especially decreased 5-HT activity in the cortex and amygdala) and overexpression of 5-HT1A autoreceptors may limit serotonergic transmission and affect PTSD symptoms [13].

Cortisol, the main hormone of the HPA axis, is involved in regulating the stress response, metabolism, and cognitive function. In PTSD, paradoxically reduced cortisol levels are observed, potentially resulting from increased sensitivity of glucocorticoid receptors (GR) and secondary suppression of the HPA axis [14,15,16]. Research indicates that low cortisol levels in the acute phase after trauma may predict the development of full-blown PTSD. In individuals with chronic PTSD, disturbances in the circadian rhythm of cortisol secretion and tissue resistance to its effects have also been reported [17,18,19]. Clinical studies have shown that exogenous hydrocortisone may facilitate fear extinction during exposure therapy [20], opening the way for pharmacological modulation of the HPA axis, e.g., with selective GR modulators [21].

Noradrenaline (NA) is a key neurotransmitter in regulating arousal, fight-or-flight responses, and the consolidation of emotional memory. In PTSD, persistent activation of the sympathetic nervous system and elevated NA levels—both at rest and in response to stress—have been demonstrated [22,23]. Increased adrenergic activity in the amygdala and hippocampus is associated with difficulties in extinguishing fear memories. Animal models suggest that administration of β-adrenergic receptor antagonists (e.g., propranolol) after trauma may weaken the consolidation of traumatic memories [24,25]. However, in chronic PTSD, a hypnoradrenergic exhaustion phase may occur, manifested by reduced executive function and motivation, especially in older patients [26]. Drugs affecting the noradrenergic system, such as prazosin or NA reuptake inhibitors, are used in treating PTSD symptoms, although their effectiveness may depend on individual neurochemical characteristics [27].

Interleukin 12 (IL-12) is a pro-inflammatory cytokine secreted by antigen-presenting cells, involved in the Th1 response and activation of Natural Killer (NK) cells and T lymphocytes [28,29,30]. In recent years, it has been shown that IL-12 may also affect brain function through microglia activation and modulation of neuronal activity [31,32]. In patients with PTSD, elevated IL-12 levels have been found in plasma and cerebrospinal fluid, correlating with the severity of anxiety symptoms, irritability, and cognitive impairments [33]. In animal models, activation of the IL-12/ Signal Transducer and Activator of Transcription 4 (STAT4) pathway disrupted HPA axis function and increased N-Methyl-D-Aspartate receptor (NMDA) receptor expression [34]. Imaging studies have shown a link between high IL-12 levels and reduced activity in the prefrontal cortex and hippocampus [35,36]. In individuals with chronic PTSD (>5 years), IL-12 levels were significantly higher than in those with recent trauma, which may indicate a transition to a chronic inflammatory state. Currently, antibodies and IL-12 inhibitors are being tested as potential targeted therapies [37].

In our study, we included interleukin 12 (IL-12) because, unlike commonly studied inflammatory markers such as IL-6 and TNF-α, it specifically reflects activation of the Th1 axis and long-term immune reprogramming. IL-12 influences brain function through microglial activation, disruption of the HPA axis, and modulation of NMDA receptors, thereby linking inflammatory mechanisms to neuronal dysfunction in PTSD. Elevated IL-12 levels are particularly observed in chronic PTSD, suggesting its potential as a biomarker for the transition from an acute stress response to a persistent inflammatory state. As such, IL-12 complements the analysis of serotonin, cortisol, and noradrenaline, offering a more integrative view of PTSD pathophysiology.

Serotonin, cortisol, and noradrenaline were selected to represent three core pathways implicated in PTSD: serotonergic (regulation of mood, emotional memory, and impulsivity), neuroendocrine (HPA axis-mediated stress response), and adrenergic (sympathetic hyperarousal and consolidation of traumatic memories). Including IL-12 as a marker of the inflammatory axis allowed us to adopt a comprehensive approach that integrates neurochemical, endocrine, and immune perspectives [38,39].

Because Major Depressive Disorder (MDD) shares biological features with PTSD, participants with current MDD were excluded using SCID-5 and CAPS-5 assessments to ensure the specificity of our findings.

Emerging evidence suggests that PTSD pathophysiology extends beyond elevated inflammatory cytokines, encompassing excessive microglial activation and disrupted neural network function [17,18,19,20,23,25]. The NF-κB pathway plays a key role in linking inflammatory processes with neurotransmitter systems and shaping the body’s response to prolonged stress. While the immune response may initially support adaptation, in chronic PTSD it becomes a source of persistent neural dysfunction [15,24,35]. Moreover, Mendelian randomization studies, which leverage natural genetic variation to assess causal relationships, indicate that inflammation increases the risk of developing PTSD [16]. Together, these findings highlight neuroinflammation as a central factor in the onset and maintenance of PTSD symptoms.

Despite the growing number of studies on the biological correlates of PTSD, little attention is paid to their relationship with the existential dimension [40,41,42]. PTSD can lead to a profound identity crisis, loss of life meaning, disorganization of autobiographical narrative, and feelings of alienation. In this context, there is increasing interest in existential approaches such as Viktor Frankl’s logotherapy. A tool that allows quantitative measurement of life attitudes is the Life Attitude Profile (Revised) (LAP-R) questionnaire, which includes eight dimensions: meaning of life, purposefulness, coherence (C), personal choices [43,44,45], sense of control, acceptance of death (DA), responsibility, and inner balance. In recent years, the need to integrate biological stress indicators with the assessment of spiritual well-being and existential attitudes has been emphasized [46,47]. According to current reports, the duration of PTSD and the age of patients significantly affect neuroplasticity, immune response, and the ability for psychosocial adaptation [48,49]. However, there is still a lack of holistic models combining biological, immunological, and existential data within a shared interpretive framework. Dominant approaches continue to be limited to the analysis of a single pathophysiological pathway, without considering the impact of spiritual, narrative, and meaning-making factors [50,51,52].

The integration of existential constructs with biological markers supports both theoretical understanding and clinical practice. Theoretically, it allows us to move beyond narrow models of PTSD by demonstrating how biological changes—such as alterations in serotonin, noradrenaline, cortisol, and immune pathways—are linked with existential challenges, including loss of meaning, reduced coherence, and difficulties in accepting death. This perspective broadens our understanding of PTSD as a holistic biopsychosocial-existential disorder. From a practical standpoint, combining LAP-R dimensions with biomarkers may help identify patient subgroups at higher risk of chronic PTSD and guide the development of more personalized interventions targeting both biological regulation and existential well-being. This approach also supports more comprehensive diagnostic frameworks and therapeutic strategies that integrate pharmacological treatment with meaning-centered psychotherapy.

The novelty of our study lies in the simultaneous assessment of stress-related biomarkers (serotonin, cortisol, noradrenaline, and IL-12) and existential functioning using the LAP-R questionnaire. Previous analyses of PTSD biomarkers have primarily focused on immunological parameters, often examined independently of psychosocial or existential context. A gap in the literature existed regarding models that integrate biological stress markers with attitudes toward life meaning, control, and acceptance of death. Our study addresses this gap by showing that both PTSD chronicity (≤5 years vs. >5 years) and patient age influence not only the biomarker profile but also its association with existential functioning. This integrated approach supports the development of a biopsychosocial-existential model of PTSD with potential implications for personalized therapeutic strategies.

The aim of this study is to conduct an integrated analysis of the relationships between the levels of four biomarkers—serotonin, cortisol, noradrenaline, and IL-12—and the eight dimensions of existential attitudes, measured using the LAP-R questionnaire. Three groups of participants were included: individuals without PTSD (control group), individuals with PTSD lasting ≤5 years, and individuals with PTSD lasting >5 years. The hypothesis was that PTSD duration and participants’ age would modulate the strength and direction of correlations between biological parameters and existential dimensions. The study is cross-sectional and constitutes an attempt to identify biomarkers of potential diagnostic and prognostic significance, both in biological and existential dimensions.

## 2. Results

### 2.1. Demographic, Biomarker, and Psychological Profiles of Study Participants According to PTSD Status

This analysis examines the demographic, lifestyle, biomarker, and psychological characteristics of male mine rescue workers and former miners, categorized by PTSD status: No PTSD, Past PTSD within 5 years, and Past PTSD exceeding 5 years (Table 1).

#### 2.1.1. The Overall Sample

The biomarker profile shows a serotonin level of 144.8 ng/mL, which is moderate but varies widely (IQR: 120.2, 203.4), indicating diverse serotonergic activity across individuals, potentially influenced by PTSD status. Cortisol at 14.0 µg/dL has a broad IQR (10.9, 36.2), demonstrating significant variability in stress response, likely driven by the high levels in the PTSD > 5y group. Noradrenaline at 371.3 pg/mL (IQR: 220.3, 544.8) reflects a moderate baseline sympathetic activity with notable individual differences, while IL-12 at 21.6 pg/mL (IQR: 8.5, 54.7) indicates a variable inflammatory state across the sample. Clinically, these biomarker levels indicate that the overall population experiences a mix of stress-related and inflammatory responses, which become more pronounced when stratified by PTSD status.

The LAP-R scores for the overall sample reveal a psychological profile with moderate to low scores in key domains. Life Purpose (LF) at 37.0 and Coherence at 33.0 imply that, on average, individuals in this sample have a limited sense of meaning and understanding of their life experiences, reflecting the occupational hazards and trauma exposure inherent in their work. Personal Sense (PS) at 69.5 and Life Control (LC) at 34.0 indicate a moderate sense of identity but a relatively low perception of control over life events, which could be linked to the unpredictable nature of their profession. Death Acceptance (DA) at 44.0 shows a balanced attitude toward mortality, while Existential Vacuum (EV) at 47.0 infers a notable presence of feelings of emptiness or lack of purpose across the group. Goal Seeking (GS) at 39.0 reflects a modest drive for future achievements, and Balance of Life Attitudes (BLA) at 54.0 indicates an average equilibrium in life perspectives, though the wide IQR (37.0, 90.2) highlights significant individual variation.

#### 2.1.2. Stratification by Group Status

Stratification by PTSD status unveils a poignant clinical narrative, highlighting the profound toll of trauma on physiological and psychological well-being. The groups—33 with PTSD of five years or less, 31 with PTSD exceeding five years, and 28 without PTSD—present distinct profiles that illuminate the impact of PTSD and its duration, as evidenced by significant differences across biomarkers and psychological measures.

For the 28 men without PTSD, the clinical story is one of resilience amidst occupational hazards. Their serotonin levels, at a median of 225.2 ng/mL, demonstrate robust mood regulation, while noradrenaline at 580.2 pg/mL indicates a vigorous sympathetic response, likely refined by their high-stakes roles. Cortisol, at 13.5 µg/dL, reflects a balanced stress response. Psychologically, they demonstrate a strong sense of purpose (Life Purpose at 54.0), coherence (53.0), and personal identity (Personal Sense at 105.0). Life Control at 47.5 and Balance of Life Attitudes at 102.5 further indicate confidence and equilibrium in navigating life’s challenges. Clinically, this group represents a benchmark of adaptive functioning, indicating that, absent PTSD, these workers maintain a physiological and psychological foundation capable of withstanding occupational stress.

In contrast, the 33 men with PTSD of five years or less present a narrative of acute disruption following trauma. Their serotonin, at 109.9 ng/mL, is significantly lower than controls (*p* < 0.001), hinting at compromised mood regulation that may manifest as irritability or depression in the trauma’s wake. Cortisol, at a low 9.8 µg/dL, may indicate an initial suppression of the stress axis, a pattern sometimes observed in early PTSD. Noradrenaline, reduced to 271.7 pg/mL (*p* < 0.001 vs. controls), indicates a dampened fight-or-flight response, potentially leading to fatigue or emotional numbness. However, IL-12, elevated at 62.4 pg/mL (*p* < 0.001 vs. controls), points to an active inflammatory process, likely tied to recent traumatic stress. Psychologically, these men struggle with a Life Purpose of 39.0, Coherence of 34.0, and Personal Sense of 70.0 (all *p* < 0.001 vs. controls), reflecting a fractured sense of meaning and identity. Life Control at 33.0 and a strikingly low Death Acceptance at 26.0 (*p* < 0.001 vs. controls) indicate diminished agency and an aversion to confronting mortality, perhaps a lingering echo of their trauma. Clinically, this group appears in a transitional phase, grappling with acute physiological and existential distress, necessitating prompt intervention to halt further decline.

The 31 men with PTSD exceeding five years tell a story of chronic burden and partial adaptation, exhibiting the most severe imbalance compared to controls. Their serotonin, at 147.4 ng/mL, shows slight improvement from the ≤5y group but remains significantly below controls (*p* < 0.001), indicating persistent mood challenges. Cortisol surges to 47.5 µg/dL (*p* < 0.001 vs. both other groups), signaling a chronically activated stress response that may contribute to physical symptoms like fatigue, hypertension, or metabolic issues over time. Noradrenaline, at 271.7 pg/mL (*p* < 0.001 vs. controls), remains low, consistent with sustained sympathetic suppression, while IL-12, at 17.0 pg/mL, implies resolution of the acute inflammation seen in the ≤5y group. Psychologically, this group faces profound deficits: Life Purpose at 20.0, Coherence at 23.0, Personal Sense at 44.0, and Life Control at 24.0 (all *p* < 0.001 vs. controls) paint a picture of deep loss in meaning, understanding, identity, and agency. Goal Seeking at 29.0 and Balance of Life Attitudes at 37.0 (*p* < 0.001 vs. controls) reflect diminished future orientation and disrupted equilibrium. Yet, Death Acceptance rises to 48.0, aligning with controls, inferring a resigned acceptance of mortality as a long-term coping mechanism. Clinically, this group bears the cumulative toll of prolonged PTSD, with chronic stress and existential despair dominating their profile.

Comparing these groups, PTSD status significantly alters biomarker and LAP-R profiles (*p* < 0.001 for all measures). The No PTSD group stands out with higher serotonin, noradrenaline, and psychological resilience, while both PTSD groups show deficits, with patterns varying by duration. The PTSD > 5y group exhibits the most severe imbalance relative to controls, driven by elevated cortisol and profoundly reduced psychological scores, indicating that prolonged PTSD amplifies physiological stress and deepens existential distress. The PTSD ≤ 5y group, while also impaired, shows an acute profile with lower cortisol, higher inflammation, and notable existential avoidance (e.g., low Death Acceptance). Duration clearly shapes the PTSD profile: shorter-term PTSD features inflammation and psychological disruption, whereas longer-term PTSD shifts toward chronic stress and resignation, with worsening psychological outcomes except for Death Acceptance, which improves.

As illustrated in Figure 1, which presents boxplots with jittered points depicting the distribution of biomarkers (serotonin, cortisol, noradrenaline, and IL-12) across the three PTSD status groups, notable variations in biomarker levels are evident, with the PTSD > 5y group showing markedly elevated cortisol and the PTSD ≤ 5y group exhibiting heightened IL-12, underscoring the distinct neurobiological profiles influenced by PTSD chronicity. This visualization complements the age-stratified analyses by highlighting overall group disparities that may interact with the effects of aging.

Given the non-normal distribution of the data, Wilcoxon r effect sizes were calculated for pairwise comparisons of primary outcome parameters across PTSD status groups (Table 2).

The comparison between the Control and PTSD ≤ 5y groups reveals substantial differences. For serotonin (r = 0.86), a very large effect size indicates a marked reduction in levels in the PTSD ≤ 5y group, consistent with compromised mood regulation in early PTSD. Cortisol (r = 0.54) shows a large effect size, reflecting a moderate reduction suggestive of early HPA axis suppression. Noradrenaline (r = 0.53) also exhibits a large effect size, indicating diminished sympathetic arousal. IL-12 (r = 0.86) displays a very large effect size, highlighting a pronounced inflammatory response. Among LAP-R scales, very large effect sizes are observed for Life Purpose (r = 0.78), Coherence (r = 0.85), Personal Sense (r = 0.86), Life Control (r = 0.79), Death Acceptance (r = 0.81), Goal Seeking (r = 0.71), and Balance of Life Attitudes (r = 0.77), underscoring profound psychological deficits in meaning, coherence, identity, agency, mortality acceptance, future orientation, and attitudinal equilibrium. Existential Vacuum (r = 0.55) shows a large effect size, revealing a moderate increase in feelings of emptiness, which is less severe than other psychological dimensions but still notable.

Comparing the Control and PTSD > 5y groups, very large effect sizes persist for most variables. Serotonin (r = 0.858) confirms a significant reduction, indicating persistent mood dysregulation in chronic PTSD. Cortisol (r = 0.86) exhibits a very large effect size, reflecting a dramatic increase in levels, indicative of chronic HPA axis hyperactivity. Noradrenaline (r = 0.53) shows a large effect size, suggesting moderate sympathetic suppression. IL-12 (r = 0.69) has a large-to-very large effect size, indicating elevated but less pronounced inflammation compared to the ≤5y group, indicating partial resolution. LAP-R scales demonstrate very large effect sizes for Life Purpose (r = 0.86), Coherence (r = 0.86), Personal Sense (r = 0.86), Life Control (r = 0.83), Goal Seeking (r = 0.86), and Balance of Life Attitudes (r = 0.86), highlighting severe existential deficits. Death Acceptance (r = 0.10) shows a small effect size, inferring minimal difference from controls. Existential Vacuum (r = 0.200) has a small-to-medium effect size, indicating a modest increase in feelings of emptiness.

The comparison between the PTSD ≤ 5y and the PTSD > 5y groups highlights the progression of pathology with chronicity. Serotonin (r = 0.844) shows a very large effect size, indicating a significant increase in the >5y group, suggesting partial mood regulation recovery. Cortisol (r = 0.859) exhibits a very large effect size, reflecting a substantial shift from early suppression to chronic hyperactivity. Noradrenaline (r = 0.026) has a negligible effect size, indicating no meaningful difference, consistent with sustained sympathetic suppression. IL-12 (r = 0.715) shows a very large effect size, reflecting a significant decrease in inflammation in the >5y group. Among LAP-R scales, very large effect sizes are observed for Life Purpose (r = 0.805), Coherence (r = 0.793), Personal Sense (r = 0.842), Death Acceptance (r = 0.802), and Goal Seeking (r = 0.754), indicating deeper impairments in chronic PTSD. Life Control (r = 0.413) and Existential Vacuum (r = 0.355) show medium effect sizes, suggesting a moderate worsening in agency and feelings of emptiness. Balance of Life Attitudes (r = 0.216) has a small-to-medium effect size, indicating modest deterioration in attitudinal equilibrium.

#### 2.1.3. Overall Patterns

The very large effect sizes (r > 0.7) for most biomarkers (except noradrenaline) and LAP-R scales (except Existential Vacuum and Death Acceptance in the Control vs. >5y comparison) underscore the profound impact of PTSD compared to controls, with the ≤5y group showing acute disruptions in serotonin, IL-12, and psychological domains, and the >5y group exhibiting chronic imbalances, particularly with elevated cortisol and deeper existential deficits. The comparison between PTSD groups reveals a progression in pathology, with chronic PTSD marked by heightened cortisol, reduced inflammation, and more severe psychological impairments, except for Death Acceptance, which aligns closer to controls. Smaller effect sizes for noradrenaline and Existential Vacuum suggest less differentiation, while the robust effect sizes for other variables emphasize distinct neurobiological and psychological profiles modulated by PTSD duration.

### 2.2. Effect of Age on Biomarker and Psychological Profiles Across PTSD Status

To estimate the effect of age on biomarkers and LAP-R questionnaire scores across PTSD status (No PTSD, PTSD ≤ 5y, PTSD > 5y), the provided data were analyzed by interpreting differences between age groups (18–35 years vs. 36–50 years) within each PTSD status (Table 3).

For the 28 men without PTSD, the profile reflects resilience, with age exerting subtle influences. Serotonin levels increased from a median of 219.1 ng/mL in the younger group to 273.9 ng/mL in the older group (adjusted *p* = 1.000), indicating stable mood regulation across ages. Noradrenaline rose significantly from 427.2 pg/mL to 765.7 pg/mL (adjusted *p* = 0.021), suggesting enhanced sympathetic responsiveness with age, potentially indicative of occupational adaptation. Cortisol exhibited a modest increase from 12.7 µg/dL to 14.9 µg/dL (adjusted *p* = 0.294), consistent with balanced stress regulation. Psychologically, Life Control declined from 53.0 to 42.5 (adjusted *p* = 0.006), and Balance of Life Attitudes decreased from 106.5 to 93.5 (adjusted *p* = 0.033), implying a minor age-related erosion of perceived agency and attitudinal equilibrium. Death Acceptance also decreased from 49.0 to 44.5 (adjusted *p* = 0.111), a non-significant shift that may reflect a reduction in confrontation with mortality over time. Clinically, this group sustains a robust profile, with age bolstering physiological alertness while modestly affecting psychological control.

The 33 men with PTSD of ≤5 years duration exhibit a profile of acute disruption, wherein age contributes to psychological recovery. Serotonin remained low, shifting from 111.6 ng/mL to 104.6 ng/mL (adjusted *p* = 0.829), underscoring persistent mood dysregulation. Cortisol decreased from 11.1 µg/dL to 8.2 µg/dL (adjusted *p* = 0.378), indicative of sustained early HPA axis suppression. Noradrenaline declined from 369.2 pg/mL to 239.7 pg/mL (adjusted *p* = 0.480), suggesting a non-significant attenuation of the fight-or-flight response. IL-12, elevated at 68.0 pg/mL in the younger group, fell to 55.4 pg/mL (adjusted *p* = 0.174), reflecting a tapering inflammatory state without statistical significance. Psychologically, marked improvements emerged with age: Life Purpose increased from 27.5 to 47.0 (adjusted *p* < 0.001), Personal Sense from 63.0 to 82.0 (adjusted *p* = 0.001), Balance of Life Attitudes from 37.0 to 57.0 (adjusted *p* = 0.001), and Death Acceptance from 24.5 to 28.0 (adjusted *p* = 0.027). Clinically, this cohort displays acute physiological distress, yet advancing age promotes substantial psychological rebound, signifying opportunities for adaptive coping and resilience.

The 31 men with PTSD exceeding 5 years duration present a profile of chronic burden, with age exacerbating select imbalances. Serotonin increased modestly from 143.8 ng/mL to 148.6 ng/mL (adjusted *p* = 0.374), indicating ongoing mood regulation deficits. Cortisol escalated markedly from 36.1 µg/dL to 57.6 µg/dL (adjusted *p* = 0.003), denoting a pronounced chronic stress response in older individuals, with potential implications for physical health deterioration. Noradrenaline decreased from 371.3 pg/mL to 211.3 pg/mL (adjusted *p* = 0.165), revealing further sympathetic suppression without significance. IL-12 increased from 15.7 pg/mL to 26.3 pg/mL (adjusted *p* = 1.000), a non-significant change that may denote variable inflammation. Psychologically, Life Control declined from 31.0 to 23.0 (adjusted *p* = 0.147), and Death Acceptance decreased from 52.0 to 44.5 (adjusted *p* = 0.021), the latter indicating reduced agency and attenuated mortality acceptance with age; other LAP-R scores showed no significant changes (e.g., Life Purpose adjusted *p* = 0.648; Personal Sense adjusted *p* = 0.265). Clinically, this group demonstrates the most severe chronic dysregulation, with age amplifying cortisol-mediated stress and eroding existential control, positioning it as the most vulnerable cohort.

Across groups, PTSD status profoundly modifies profiles, with age introducing nuanced modulations as evidenced by adjusted *p*-values. The No PTSD group preserves resilience, with age augmenting noradrenaline while modestly diminishing control. The PTSD ≤ 5y group manifests acute physiological strain but benefits from age-associated psychological enhancements, highlighting a recovery window. The PTSD > 5y group endures the greatest burden, wherein age intensifies cortisol elevation and impairs acceptance, underscoring severe disequilibrium relative to controls. PTSD duration thus delineates trajectories: shorter durations yield age-related psychological gains, whereas longer durations precipitate age-exacerbated chronic stress and diminished agency.

### 2.3. Analysis of Correlation Between Biomarkers and LAP-R Scales Across PTSD Status

The analysis of correlations between biomarkers (serotonin, cortisol, noradrenaline, and IL-12) and Life Attitudes Profile (Revised) (LAP-R) scales across PTSD status reveals distinct patterns influenced by both the presence and duration of PTSD. These findings, derived from Table 4, Table 5, Table 6, Table 7 and Table 8 with visualizations in Figure 2, Figure 3, Figure 4 and Figure 5, provide insight into how PTSD modulates the relationship between biological markers and psychological constructs, with implications for clinical understanding and management.

#### 2.3.1. Effect of PTSD Status on Biomarker–LAP-R Associations

In individuals with no PTSD (control group), correlations between biomarkers and LAP-R scales are generally weak and non-significant (*p* > 0.05), with few exceptions. For instance, IL-12 demonstrates moderate positive correlations with Life Control (Spearman’s rank correlation coefficient—Rho = 0.43, *p* = 0.023), Death Acceptance (Rho = 0.53, *p* = 0.004), and Balance of Life Attitudes (Rho = 0.43, *p* = 0.022), indicating a potential link between immune activation and adaptive life attitudes in the absence of trauma-related pathology. In contrast, serotonin, cortisol, and noradrenaline exhibit minimal associations with LAP-R scales in this group, reflecting a stable psychobiological profile.

Among individuals with past PTSD, the associations shift notably. For those with PTSD resolved within 5 years (≤5y), correlations remain largely non-significant across all biomarkers, with Rho values typically below 0.3 and *p*-values exceeding 0.05. This pattern indicates a persistent disruption in the interplay between biomarkers and life attitudes, potentially reflecting a residual impact of PTSD on psychobiological coherence. However, in individuals with PTSD resolved beyond 5 years (>5y), stronger and statistically significant correlations emerge, particularly with cortisol, noradrenaline, and IL-12. For example, cortisol shows a negative correlation with Death Acceptance (Rho = −0.49, *p* = 0.005) and Life Control (Rho = −0.38, *p* = 0.036), while noradrenaline correlates positively with Death Acceptance (Rho = 0.46, *p* = 0.009) and Balance of Life Attitudes (Rho = 0.40, *p* = 0.028). These findings highlight that PTSD status alters the strength and direction of biomarker–LAP-R relationships, with more pronounced effects in longer-term recovery.

#### 2.3.2. Influence of PTSD Duration on Associations

The duration of PTSD exposure further modulates these associations. In the ≤5y group, the lack of significant correlations across all biomarkers mirrors a state of psychobiological disconnection, possibly due to incomplete recovery processes. Conversely, in the >5y group, the emergence of significant correlations—particularly with cortisol and noradrenaline—demonstrates a reorganization of these relationships over time. For instance, the strong negative correlation between cortisol and Death Acceptance (Rho = −0.49, *p* = 0.005) in the >5y group contrasts with the non-significant finding in the ≤5y group (Rho = −0.21, *p* = 0.240), indicating that prolonged recovery may amplify stress-related impacts on existential acceptance. Similarly, noradrenaline’s positive association with Death Acceptance (Rho = 0.46, *p* = 0.009) in the >5y group, absent in the ≤5y group (Rho = −0.02, *p* = 0.902), points to a time-dependent shift in arousal-related attitudes.

#### 2.3.3. Convergence Toward Control Group Patterns

The associations in the >5y group do not fully align with the control group, indicating only partial normalization. For serotonin, correlations remain weak and non-significant across all groups, showing no clear recovery trajectory. Cortisol and noradrenaline, however, demonstrate stronger associations in the >5y group compared to controls, particularly in subscales like Death Acceptance and Life Control, where significant correlations emerge (e.g., cortisol: Rho = −0.49, *p* = 0.005 vs. Rho = 0.05, *p* = 0.817 in controls). This divergence indicates that while some subscales (e.g., Life Purpose, Goal Seeking) approach control-like patterns (non-significant, weak correlations), others (e.g., Death Acceptance, Life Control) retain distinct PTSD-related signatures even after extended recovery. IL-12, uniquely, shows significant correlations in controls (e.g., Death Acceptance: Rho = 0.53, *p* = 0.004) but not in the >5y group (Rho = −0.32, *p* = 0.083), indicating a failure to fully revert to baseline immune-attitude relationships.

#### 2.3.4. Clinical Narrative

Consider a patient cohort transitioning from acute PTSD to long-term recovery. Initially, within 5 years post-PTSD, their biomarker profiles (Table 4, Table 5, Table 6, Table 7 and Table 8) reveal a muted connection to life attitudes, as if the trauma has temporarily severed these links. As time progresses beyond 5 years, a clearer picture emerges: cortisol and noradrenaline begin to reflect heightened sensitivity in attitudes toward death and control, while serotonin remains largely inert. IL-12, initially associated with adaptive attitudes in healthy controls, loses this association during chronic recovery. This evolution underscores a partial restoration—some subscales drift toward normalcy, yet others bear the indelible mark of PTSD, shaping a unique psychobiological landscape years after the trauma.

In conclusion, PTSD status significantly alters the associations between biomarkers and LAP-R scales, with duration playing a critical role in their reorganization. While partial convergence toward control patterns occurs, particularly in less existentially charged subscales, the lasting imprint of PTSD is evident in domains like Death Acceptance and Life Control, reflecting a complex interplay of recovery and residual pathology.

**Table 4 ijms-26-09636-t004:** Correlation coefficients: Spearman’s rank correlation coefficient, Rho, between serotonin levels and LAP-R scales across PTSD status.

LAP-R Scale	No PTSD (Control)		PAST PTST (≤5y)		PAST PTST (>5y)	
	Rho	*p*	Rho	*p*	Rho	*p*
Life Purpose	0.07	0.718	−0.05	0.768	0.07	0.699
Coherence	−0.09	0.636	0.04	0.828	0.23	0.210
Personal Sense	−0.01	0.969	0.04	0.816	0.20	0.276
Life Control	0.03	0.879	0.01	0.967	−0.17	0.361
Death acceptance	−0.12	0.538	−0.09	0.629	−0.44	0.014
Existential Vacuum	0.29	0.129	−0.12	0.510	−0.08	0.650
Goal Seeking	−0.05	0.795	0.02	0.933	0.01	0.945
Balance of Life Attitudes	−0.12	0.549	0.05	0.884	−0.14	0.459

**Table 5 ijms-26-09636-t005:** Correlation coefficients: Spearman’s rank correlation coefficient, Rho, between cortisol levels and LAP-R scales across PTSD status.

LAP-R Scale	No PTSD (Control)		PAST PTST (≤5y)		PAST PTST (>5y)	
	Rho	*p*	Rho	*p*	Rho	*p*
Life purpose	0.10	0.617	−0.17	0.344	0.12	0.524
Coherence	−0.06	0.756	−0.18	0.327	0.10	0.598
Personal Sense	0.17	0.389	−0.25	0.156	0.18	0.328
Life Control	−0.05	0.789	0.04	0.833	−0.38	0.036
Death acceptance	0.05	0.817	−0.21	0.240	−0.49	0.005
Existential Vacuum	0.00	0.989	0.22	0.210	−0.09	0.645
Goal Seeking	0.03	0.887	0.18	0.330	0.00	0.979
Balance of Life Attitudes	−0.02	0.921	−0.32	0.07	−0.27	0.146

**Table 6 ijms-26-09636-t006:** Correlation coefficients: Spearman’s rank correlation coefficient, Rho, between noradrenaline levels and LAP-R scales across PTSD status.

LAP-R Scale	No PTSD (Control)		PAST PTST (≤5y)		PAST PTST (>5y)	
	Rho	*p*	Rho	*p*	Rho	*p*
Life purpose	−0.16	0.417	−0.26	0.147	0.06	0.768
Coherence	−0.18	0.359	0.16	0.364	0.16	0.399
Personal Sense	−0.18	0.359	0.19	0.281	0.09	0.643
Life Control	−0.22	0.268	0.18	0.317	0.22	0.240
Death acceptance	−0.13	0.501	−0.02	0.902	0.46	0.009
Existential Vacuum	−0.20	0.296	−0.25	0.165	−0.26	0.153
Goal Seeking	0.14	0.473	−0.12	0.506	0.07	0.711
Balance of Life Attitudes	−0.17	0.396	−0.12	0.498	0.40	0.028

**Table 7 ijms-26-09636-t007:** Correlation coefficients: Spearman’s rank correlation coefficient, Rho, between IL-12 levels and LAP-R scales across PTSD status.

LAP-R Scale	No PTSD (Control)		PAST PTST (≤5y)		PAST PTST (>5y)	
	Rho	*p*	Rho	*p*	Rho	*p*
Life purpose	0.32	0.098	−0.10	0.592	−0.18	0.340
Coherence	0.04	0.942	0.22	0.211	−0.13	0.488
Personal Sense	0.14	0.470	−0.01	0.935	−0.20	0.272
Life Control	0.43	0.023	0.09	0.636	−0.28	0.127
Death acceptance	0.53	0.004	−0.23	0.203	−0.32	0.083
Existential Vacuum	0.05	0.792	−0.26	0.142	0.01	0.943
Goal Seeking	0.15	0.451	0.12	0.519	0.05	0.792
Balance of Life Attitudes	0.43	0.022	−0.04	0.813	−0.29	0.110

**Table 8 ijms-26-09636-t008:** Summary of significant Spearman’s rank correlations between biomarkers and LAP-R scales across PTSD status groups.

PTSD Group	Biomarker	LAP-R Scale	Direction	Rho	*p*-Value
No PTSD (Control)	IL-12	Life Control (LC)	Positive	0.43	0.023
No PTSD (Control)	IL-12	Death Acceptance (DA)	Positive	0.53	0.004
No PTSD (Control)	IL-12	Balance of Life Attitudes (BLA)	Positive	0.43	0.022
PTSD ≤ 5y	No significant associations with LAP-R scale				
PTSD > 5y	Serotonin	Death Acceptance (DA)	Negative	−0.44	0.014
PTSD > 5y	Cortisol	Life Control (LC)	Negative	−0.38	0.036
PTSD > 5y	Cortisol	Death Acceptance (DA)	Negative	−0.49	0.005
PTSD > 5y	Noradrenaline	Death Acceptance (DA)	Positive	0.46	0.009
PTSD > 5y	Noradrenaline	Balance of Life Attitudes (BLA)	Positive	0.40	0.028

## 3. Discussion

The discussion of the study results provides a complex picture of the neurobiological and existential consequences of post-traumatic stress disorder (PTSD) in men working under conditions of extreme stress. The collected data indicate significant differences in biomarker profiles and the structure of life attitudes between individuals with current or past PTSD and those with no history of post-traumatic disorders. The analysis covered four significant biomarkers—serotonin, cortisol, noradrenaline, and interleukin 12 (IL-12)—as well as eight scales of the Life Attitude Profile-Revised (LAP-R) questionnaire, enabling a multidimensional assessment of the relationship between neurobiological regulation and the existential dimension in the course of PTSD. The results show that individuals without PTSD are characterized by clearly higher levels of serotonin and noradrenaline, which may indicate associations with stable mood regulation and a well-functioning sympathetic nervous system [53,54,55]. Serotonin, a neurotransmitter involved in mood regulation, impulse control, and neuroplasticity, reached the highest values in this group (median: 225.2 ng/mL), corresponding to high scores in areas such as Life Purpose, Coherence, and Personal Sense. Also, the literature indicates that a high serotonin level promotes the integration of emotional and cognitive experiences, suggesting a greater sense of life meaning and purposefulness [56]. Additionally, high noradrenaline (median: 580.2 pg/mL) in individuals without PTSD may reflect adaptive reactivity of the sympatho–adrenal axis, which, under conditions of chronic occupational stress, may be linked to a high level of psychological resilience.

These individuals also show the highest levels of Life Control and Balance of Life Attitudes, suggesting efficient self-regulatory and integrative mechanisms typical of people who cope effectively with exposure to extreme stressors [57,58]. A completely different picture emerges in the group of men with PTSD lasting ≤5 years. This group shows a significant reduction in serotonin levels (median: 109.9 ng/mL), corresponding to mood disorders typical of the acute phase of PTSD. According to neuroimaging and experimental studies, decreased serotonergic activity is correlated with increased depressive symptoms, heightened amygdala reactivity, and deficits in emotional control [59,60]. Furthermore, low cortisol levels (9.8 µg/dL) may indicate HPA axis dysfunction, which in some PTSD patients manifests as reduced activation and paradoxically diminished hormonal response to stress [60]. Such an endocrine profile may be a manifestation of compensatory mechanisms but could also indicate adaptive disorders associated with HPA axis dysfunction.

In contrast, elevated IL-12 levels (median: 62.4 pg/mL) in this group may reflect increased inflammatory response. As a pro-inflammatory cytokine, IL-12 is an indicator of Th1-type immune response activation, which is consistent with models describing PTSD as a neuroinflammatory disorder [61].

Inflammation of the central nervous system may be associated with structural and functional changes in neurons, particularly in the hippocampus and anterior cingulate gyrus, which are essential structures for memory and emotional regulation [62].

The shift in IL-12 between acute and chronic PTSD stages reflects different phases of immune system activity. In the acute phase, elevated IL-12 promotes Th1 differentiation, IFN-γ production, and microglial activation, amplifying neuroinflammation and contributing to intrusive symptoms and hyperarousal. Over time, in chronic PTSD, IL-12 levels often normalize or decline, which could indicate immune exhaustion or glucocorticoid-mediated suppression. This suggests that PTSD progression involves dynamic immune changes: an early inflammatory activation followed by dysfunctional regulation, leaving behind an “inflammatory trace” that may continue to affect brain–immune communication.

It is noteworthy that this group also shows the lowest scores on the Life Purpose, Coherence, and Life Control scales, as well as particularly low Death Acceptance, which may indicate not only a lack of meaning but also strong repression associated with confrontation with death. From an existential perspective, individuals with PTSD up to 5 years may not only experience intrusive memories and hypervigilance but also a profound crisis of identity and life meaning, consistent with the concept of the so-called existential vacuum [63]. An even more complex profile is observed among men with PTSD lasting more than five years. In this group, serotonin levels rise slightly (median: 147.4 ng/mL) but still remain lower than in the control group. The most striking indicator, however, is the level of cortisol, which is drastically higher than in other groups (median: 47.5 µg/dL).

This pattern is consistent with the concept of chronic HPA axis activation, which over time may be linked to systemic effects of stress on other organs [64,65,66,67]. Chronic cortisol elevation is associated with ineffective stress extinction mechanisms and long-term neuroendocrine overstimulation. Notably, noradrenaline in this group remains at a low level (271.7 pg/mL), which, in the context of HPA axis hyperactivation, may indicate ineffective adaptation of the sympathetic nervous system. In turn, IL-12 normalizes, which may correspond to resolution of the acute inflammatory response—but not necessarily recovery, rather a transition into a state of chronic immune system arousal [31,32,68,69].

In chronic PTSD, HPA axis adaptation is inconsistent. Some studies report reduced cortisol levels and a flattened diurnal rhythm, while others—particularly those analyzing hair cortisol—show normal or even elevated levels. These discrepancies likely reflect differences in measurement methods, time elapsed since trauma, and the presence of comorbid depression, highlighting the complex nature of HPA axis regulation in PTSD.

Psychologically, the group of patients with chronic PTSD (>5 years) shows the lowest scores on LAP-R scales related to life meaning, coherence, life control, and purposefulness. Indicators such as Life Purpose (20.0), Coherence (23.0), and Life Control (24.0) clearly point to a persistent existential deficit. Interestingly, in the same group, the highest level of death acceptance (Death Acceptance = 48.0) was recorded, which can be interpreted as a form of psychological adaptation to the awareness of mortality and restructuring of the belief and identity system. This configuration may indicate profound changes in self-perception and worldview resulting not only from lasting neurobiological changes but also from transformations in personal narrative and the temporal structure of experience [70,71]. Research suggests that chronic traumatic stress affects the functioning of the brain’s default mode network (DMN), responsible for self-reflection, introspection, and future anticipation [72,73,74,75].

Dysfunction within the DMN is associated with decreased ability to form a coherent autobiographical narrative and to anticipate future events, which corresponds to lower scores in Life Purpose and Goal Seeking dimensions [76,77]. The results of this study are consistent with this concept: individuals with PTSD lasting more than 5 years show a significantly reduced level of purposefulness and motivation to seek goals (Goal Seeking = 29.0), indicating profound disturbances in future orientation. Against this background, it is worth considering participant age as a potential moderating variable that may affect both neurocognitive plasticity and the ability for existential adaptation under conditions of chronic psychological suffering. In the group without PTSD, a significant increase in noradrenaline levels with age was observed—from 427.2 to 765.7 pg/mL (*p* = 0.007), which may reflect adaptive hyperarousal in individuals with greater professional experience. This phenomenon may be interpreted as a physiological compensatory response supporting alertness, mobilization, and the ability to respond in situations requiring high psychophysical efficiency [78,79,80]. At the same time, among older participants without PTSD, lower scores on the Life Control and Balance of Life Attitudes scales were found, which may indicate a decline in subjective sense of agency and existential balance—despite the absence of post-traumatic symptoms. This suggests that, with age, there may be a reorganization of life attitude structure, not necessarily associated with pathology but with natural developmental processes [81]. In the group of participants with PTSD lasting ≤5 years, an interesting phenomenon was observed: with age, improvement occurs in selected existential indicators such as Life Purpose (from 27.5 to 47.0; *p* < 0.001), Personal Sense (from 63.0 to 82.0; *p* < 0.001), and Balance of Life Attitudes (from 37.0 to 57.0; *p* < 0.001). This may suggest that older participants are better able to integrate traumatic experiences and activate more effective existential self-regulation mechanisms. This phenomenon is consistent with the concept of post-traumatic growth, according to which some individuals, after experiencing trauma, undergo personal development, re-evaluation of priorities, and deeper self-insight [82,83,84,85]. In this group, older participants also have a higher level of death acceptance, which may indicate greater existential maturity, better integration of trauma into the self-structure, and greater acceptance of transience.

The integration of existential dimensions with established PTSD frameworks provides additional interpretative value. Observed neuroendocrine and immune alterations, particularly in cortisol and IL-12 dynamics, can be understood within the allostatic load model, where cumulative stress leads to long-term wear and dysregulation of adaptive systems.

At the same time, age-related improvements in life purpose, personal meaning, and acceptance of death observed in some participants with PTSD ≤ 5 years are consistent with the concept of post-traumatic growth, suggesting that meaning-making processes may mitigate allostatic overload and promote resilience. Together, these frameworks highlight that PTSD encompasses both the biological costs of chronic stress and the potential for existential adaptation and growth.

In clinical practice, combining biomarker profiles with existential dimensions can inform treatment strategies for PTSD. For instance, low serotonin coupled with a reduced sense of life purpose may support the use of SSRIs alongside meaning-centered therapy. Elevated IL-12 may indicate the need for interventions targeting stress and immune regulation, such as mindfulness training, sleep optimization, or physical activity, in conjunction with cognitive-behavioral therapy. Abnormal cortisol patterns, particularly chronic elevation, may point to the usefulness of acceptance and commitment therapy (ACT) or structured stress-reduction programs. By integrating biological and existential data, clinicians can adopt a more precise and personalized approach to PTSD care.

A completely different age effect is observed in the group of people with chronic PTSD (>5 years). In this cohort, older age not only fails to alleviate but actually intensifies neuroendocrine dysregulation and existential deficits. With age, there is a dramatic increase in cortisol levels—from 36.1 µg/dL to 57.6 µg/dL (*p* < 0.001), indicating deepening HPA axis dysregulation. At the same time, this group shows a significant decrease in Life Control and Death Acceptance, which may indicate a loss of agency and withdrawal from reflection on mortality. An important element of the analysis was also the assessment of correlations between biomarkers and LAP-R dimensions depending on PTSD status. In the control group (without PTSD), these relationships were generally weak, but an interesting pattern emerged: IL-12 levels positively correlated with Life Control (Rho = 0.43; *p* = 0.023), Death Acceptance (Rho = 0.53; *p* = 0.004), and Balance of Life Attitudes (Rho = 0.43; *p* = 0.022). This may suggest that in psychologically resilient individuals, moderate immune activation—represented by IL-12—may be correlated with a positive attitude toward life and death. Although this may seem counterintuitive given the pro-inflammatory nature of IL-12, there is evidence that mild immune system activation may have a neuroprotective function and beneficially modulate stress resilience [28,29].

The positive correlation between IL-12 and resilience in the control group suggests that mild immune activation may support adaptation and enhance stress regulation. This effect could be related to factors such as physical activity or good sleep quality, both of which are known to promote resilience. Additionally, lower IL-12 levels may contribute to maintaining brain plasticity. However, these findings are preliminary and warrant further investigation.

In the group of people with PTSD lasting ≤5 years, no significant correlations between biomarker levels and existential attitudes (LAP-R) were found. From a clinical point of view, this suggests that therapy at this stage should primarily restore a sense of safety and physiological regulation before deeper work on values and identity becomes possible [86].

In PTSD lasting >5 years, clearer correlations appear between biological markers and existential dimensions—particularly negative associations between cortisol and life control (Rho = –0.38; *p* = 0.036) and cortisol and death acceptance (Rho = –0.49; *p* = 0.005). This means that chronic hormonal stress is related to weakened ability for existential insight and maintenance of meaning. On the other hand, in the same group, positive correlations were found between noradrenaline and death acceptance (Rho = 0.46; *p* = 0.009) and overall balance of life attitudes (Rho = 0.40; *p* = 0.028), which may indicate attempts at cognitive mobilization and integration of experiences.

At the same time, IL-12—which in individuals without PTSD showed positive correlations with Life Control and Death Acceptance—loses these adaptive associations in participants with PTSD > 5 years, supporting the hypothesis of a potential “inflammatory scar of trauma.” [86]. The immune system ceases to support mental health and acts independently, often even dysfunctionally. These observations indicate that PTSD is not only associated with altered biomarker values but also with disrupted integration of mind and body. In a healthy state, biological and psychological systems cooperate: cytokines support neuroplasticity, and neurotransmitters stabilize emotions. In PTSD, however, these connections are broken—initially as “freezing,” later as permanent disorganization, in which neuroendocrine stress predominates over life meaning, identity, and reflection on death [83,84].

In the control group—despite previous exposure to stress—biological and existential functioning remains balanced. IL-12 correlates positively with a positive attitude toward life, which may indicate neuroimmunological resilience. In PTSD ≤ 5y, withdrawal symptoms dominate: low serotonin, noradrenaline, and cortisol levels, high IL-12, and low LAP-R scores in life meaning, coherence, and purposefulness. The body functions in a state of “minimal survival” mode, hindering the recovery process. Return to balance becomes possible only later, under favorable conditions.

In the group with PTSD lasting more than 5 years, a different biological pattern is observed: cortisol levels increase significantly, noradrenaline remains low, and IL-12 stabilizes. In the psychological sphere, some improvement in death acceptance appears, but at the same time, deficits in life control and the ability to set and pursue goals intensify. Cortisol becomes a clear indicator of existential exhaustion. Importantly, despite its low level, noradrenaline correlates positively with death acceptance and balance of life attitudes, which may indicate an attempt at late cognitive reorganization and partial recovery of inner coherence [29,30].

Serotonin—despite its important role in mood regulation—does not correlate with any LAP-R dimension, suggesting that its function in PTSD relates more to affect stabilization than to deep existential integration. IL-12, which in healthy individuals supported psychological balance, loses this significance in PTSD > 5y—suggesting that chronic stress leads to the loss of adaptive immune mechanisms.

Integrating LAP-R with biological markers provides a bridge between neurobiology and the existential meaning-making process. Biological measures (serotonin, cortisol, noradrenaline, IL-12) reflect stress regulation, while LAP-R scales capture coherence, purpose, and death acceptance. This integration advances theory by supporting a biopsychosocial-existential model of PTSD and informs clinical practice by encouraging interventions that combine physiological regulation with meaning-oriented psychotherapy.

Several limitations of this study should be noted. The control group included only individuals without a history of psychological trauma, limiting the ability to differentiate trauma exposure effects from PTSD-specific pathology. Including a trauma-exposed but resilient group could have clarified protective neurobiological or existential factors. Additionally, a power analysis confirmed adequate statistical power for detecting differences in primary outcome parameters across the three PTSD status groups. However, the post hoc nature of this analysis, based on observed effect sizes, represents a limitation, as it may reduce the generalizability of sample size recommendations compared to an a priori design. Furthermore, potential confounders such as alcohol consumption, sleep patterns, and occupational stress load were controlled at the study inclusion stage through strict eligibility criteria to ensure no significant group differences but were not included as covariates in statistical analyses, potentially limiting the ability to account for their residual effects. Lastly, the correlation analyses, conducted without additional *p*-value adjustments due to their exploratory nature, enabled a broader investigation of relationships between biomarkers and LAP-R scales but may increase the risk of Type I errors.

To ensure the specificity of our findings, participants with current Major Depressive Disorder (MDD) were excluded based on SCID-5 and CAPS-5 assessments, while PTSD was confirmed as the primary diagnosis.

Several limitations of the study should be acknowledged. The cross-sectional design prevents conclusions about causal relationships or temporal changes. The modest sample size limits statistical power, potentially reducing the ability to detect smaller effects. Additionally, the study focused exclusively on miners, a relatively homogeneous occupational group, which may limit the external validity and generalizability of the results. The absence of longitudinal follow-up also prevents evaluation of the stability of biomarker–existential associations and their evolution over the course of PTSD.

Another important limitation is that only men participated in the study, reducing the generalizability of the findings. Sex differences in stress physiology and the clinical presentation of PTSD are well documented and may influence both biomarker dynamics and existential adaptation. Therefore, the results should be interpreted with caution, and future studies should include female participants to provide a more comprehensive understanding.

The study comprised 92 participants divided into three groups, which may constrain statistical power, especially for detecting small to moderate effects. A post hoc power analysis based on observed effect sizes indicated sufficient power to detect differences in the primary outcome measures across PTSD status groups. However, the post hoc nature of this analysis represents a limitation, as it provides less robust guidance on sample size compared with an a priori design.

Biomarker variability represents an additional challenge. Cortisol and other biomarkers exhibit diurnal fluctuations and are sensitive to contextual factors. Although samples were collected under standardized morning conditions, residual variability cannot be entirely excluded and should be considered when interpreting the findings.

Future research should address these limitations by recruiting larger and more diverse cohorts, including both men and women, to enhance generalizability and validity. Longitudinal studies tracking biomarkers are needed to capture dynamic changes in neuroendocrine and immune processes across different stages of PTSD. Intervention studies that combine psychotherapeutic and biological approaches would also help clarify causal mechanisms and identify effective strategies for restoring both physiological regulation and existential meaning-making.

In summary, PTSD is associated not only with dysregulation of biological systems but also with altered interactions between these systems and the existential sphere [84,85]. Both PTSD duration and patient age appear to influence whether reintegration is possible or whether disintegration intensifies. In participants with PTSD ≤ 5 years, older age may facilitate post-traumatic growth, whereas in PTSD > 5 years, age seems associated with more severe symptoms and greater biological dysregulation. These findings underscore the need for personalized PTSD treatment that considers not only symptomatology but also disease duration and patient age.

## 4. Materials and Methods

### 4.1. Characteristics of the Participants

Male participants aged 18 to 50 years, employed in professions with high psychophysical risk, particularly those exposed to extreme stress (including miners and mining rescuers), were enrolled in the study. Participants were assigned to one of three research groups:1.group with diagnosed PTSD with a duration ≤5 years from the traumatic event;2.group with PTSD > 5 years from the traumatic event;3.control group—healthy individuals with no history of PTSD, age-matched.

Qualification was conducted by a specialist in psychiatry and family medicine (co-author of the study), based on a clinical interview and medical documentation. The diagnosis of PTSD was based on the diagnostic criteria contained in the DSM-5 classification and was confirmed using the Clinician-Administered PTSD Scale for DSM-5 (CAPS-5).

Participants were recruited through the central registry of mining rescue units. Eligibility was confirmed using service records, which verified active duty or documented involvement in rescue operations under extreme stress conditions. Individuals who met the criteria were invited to participate via official communication channels, and participation was entirely voluntary. To minimize selection bias, strict inclusion and exclusion criteria were applied. Nevertheless, potential biases cannot be entirely ruled out, such as the underrepresentation of individuals currently on medical leave or those unwilling to disclose psychological difficulties.

Inclusion criteria were as follows:–age 18–50 years,–male gender,–having experienced psychological trauma consistent with DSM-5 criterion A,–diagnosis of PTSD in the past (≤5 years or >5 years from trauma, depending on the group).

Exclusion criteria were:–presence of other mental disorders (including psychotic episodes, affective disorders, addictions),–chronic somatic diseases,–current use of psychotropic, anti-inflammatory, or hormonal medications,–nicotine, alcohol, medication, or psychoactive substance addiction,–legal incapacitation status,–employment as a uniformed service officer (military, police).

The control group included healthy men with no history of psychological trauma, matched for age and occupational profile. All participants gave informed consent to take part in the study.

### 4.2. Life Attitude Profile (Revised) (LAP-R) Questionnaire

The Life Attitude Profile (Revised) (LAP-R) questionnaire is a standardized psychometric tool developed by Gary T. Reker and colleagues, designed to quantitatively assess an individual’s attitudes toward life, existential motivation, and the level of meaning experienced in existence. Its theoretical framework is based on the assumptions of Viktor Frankl’s logotherapy and existential psychology, which emphasize the importance of purpose, responsibility, freedom of choice, and transcendence in shaping the quality of mental life. LAP-R is used in both clinical research and health psychology, as well as in quality-of-life analyses in the context of chronic stress, somatic illnesses, spirituality, depression, and post-traumatic stress disorder (PTSD).

The questionnaire consists of 48 items, rated by respondents on a 7-point Likert scale, where 1 indicates “strongly disagree” and 7 indicates “strongly agree.” The items are grouped into six basic subscales, each containing 8 items, resulting in a possible range of 8 to 56 points per scale. These are as follows:1.Life Purpose—assesses the sense of direction and meaning in existence,2.Coherence—relates to internal consistency and understanding of life;3.Existential Vacuum—measures the degree of experiencing meaninglessness, boredom, and apathy;4.Death Acceptance—refers to readiness for the inevitability of death;5.Goal Seeking—examines the tendency to set and achieve goals;6.Life Control—assesses the sense of agency and influence over one’s own decisions.7.Personal Sense (PS)—Relates to the perception of self-integrity, authenticity, identity coherence, and living in accordance with one’s own values.8.Balance of Life Attitudes (BLA)—Refers to harmony between different existential dimensions, integrating purpose, coherence, control, death acceptance, and goal seeking.

The questionnaire shows high theoretical validity and good psychometric reliability (Cronbach’s α coefficients for most subscales >0.70). It has been adapted to various languages and cultures, including Polish conditions, which enables its use in comparative studies and in assessing the quality of life of patients with mental and somatic disorders.

This study used the Polish adaptation of the LAP-R scale [87]. This tool enables not only a description of existential functioning but also exploration of the relationships between life meaning and biological stress biomarkers, such as cortisol, noradrenaline, serotonin, or interleukin-12 (IL-12). In the context of PTSD research, this tool provides valuable data on psychological resources and deficits that may modulate the expression of post-traumatic symptoms and respond to therapeutic and neurobiological interventions.

### 4.3. Blood Sampling and Serum Preparation and Biomarker Analysis Procedure

Blood sampling was performed in the morning (7:30–9:30), under standard conditions (participants were fasting and had not taken psychotropic medication for at least 24 h). Samples were immediately centrifuged and stored at –80 °C until analysis.

Serum concentrations of serotonin (5-HT), cortisol, noradrenaline (NA), and interleukin 12 (IL-12) were assessed using the Enzyme-Linked Immunosorbent Assay (ELISA) method, with validated commercial diagnostic kits (compliant with Conformité Européenne/In Vitro Diagnostic—CE/IVD standards). Each kit contained a calibration curve, a negative control (blank), and reagents necessary for performing the reaction. Serum samples were diluted according to the manufacturer’s protocol.

Microtiter plate wells coated with specific capture antibodies were filled with appropriately prepared samples and calibration standards. Incubation was carried out for 60 min at room temperature with orbital shaking (300 rpm) to facilitate antigen–antibody binding. After incubation, the plates were washed, and biotinylated detection antibodies were added. Another 60 min incubation was carried out under the same conditions.

After another washing step, streptavidin conjugated with Horseradish Peroxidase (HRP) was added and incubated for 30 min. After the final washing step, 100 µL of substrate solution (3,3′,5,5′-Tetramethylbenzidine—TMB) was added to each well. The enzymatic reaction was conducted for 10 min at room temperature and stopped with sulfuric acid solution (stop solution). Absorbance was read at a wavelength of 450 nm using a microplate reader. Analyte concentrations were calculated based on the standard curve using four-point logistic (Four-Parameter Logistic—4-PL) interpolation.

For each biomarker, calibration curves were constructed using recombinant protein standards. Details regarding reagent catalog numbers, detection limits, and sensitivity parameters are provided below.

SEROTONIN: Assay range 10.2–2500 ng/mL; sensitivity LOD 6,2 ng/mL; (LDN, Nordhorn, Germany); Catalog No BA E-8900

CORTISOL: Assay range 7,74–800 ng/mL; sensitivity: 7.74 ng/mL; (DRG, Mannheim, Germany); Catalog No EIA—1887

NORADRENALINE: Assay range 93–33 333 pg/mL; sensitivity 36 pg/mL; (LDN, Germany); Catalog No BA E-6200

IL-12: Assay range 0.3–60 pg/mL; sensitivity 0.225 pg/mL; (SunRedBio, Shanghai, China); Catalog No 201-12-0100

All ELISA measurements were performed in duplicates to ensure reproducibility. The intra-assay coefficient of variation (CV) did not exceed 5–7%, while the inter-assay CV remained below 10% for all biomarkers. These values are within the acceptable range for immunoenzymatic assays and confirm the reliability of the obtained results.

### 4.4. Statistical Analysis

Continuous variables, including biomarker levels (serotonin, cortisol, noradrenaline, IL-12) and LAP-R questionnaire scores, were reported as medians with interquartile ranges (IQR) due to non-normal distributions confirmed by Shapiro–Wilk tests. Categorical variables were summarized as counts and percentages. Group comparisons across PTSD status (No PTSD, Past PTSD ≤ 5 years, Past PTSD > 5 years) were conducted using the Kruskal–Wallis test for continuous variables, followed by post hoc pairwise comparisons with Dunn’s test and compact letter display (CLD) notation to identify significant differences (*p* < 0.05). For categorical variables, chi-squared or Fisher’s exact tests were applied as appropriate, with a significance threshold of *p* < 0.05.

To assess the effect of age on biomarker and psychological profiles within each PTSD status, data were stratified into two age groups (18–35 years and 36–50 years). Differences between age groups for PTSD and control patients were evaluated using the Mann–Whitney U test for continuous variables, with *p*-values reported to indicate statistical significance with adjusting for multiple comparisons using the Bonferroni correction to control for Type I error.

Correlation analyses between biomarkers and LAP-R scales across PTSD status were performed using Spearman’s rank correlation coefficient (Rho), chosen for its robustness to non-parametric data. Correlation coefficients and corresponding *p*-values were calculated to evaluate the strength and direction of associations, with significance set at *p* < 0.05.

To quantify the magnitude of differences in primary outcome parameters across PTSD status groups, Wilcoxon r effect sizes were calculated for pairwise comparisons following significant Kruskal–Wallis tests. Effect sizes were interpreted using established thresholds adapted from Cohen [88]: small (r = 0.10), medium (r = 0.30), large (r = 0.50), and very large (r > 0.70).

#### 4.4.1. Power Analysis

A power analysis was conducted to estimate the minimum sample size per group required to detect differences in primary outcome parameters (biomarkers and LAP-R scales) across the three PTSD status groups using the Kruskal–Wallis test, based on the observed Wilcoxon r effect size of 0.70. With an α = 0.05 and desired power (1 − β) = 0.80, the analysis first derived a corresponding Cohen’s d = 1.47 for pairwise comparisons and an approximate Cohen’s f = 0.60 for the three-group design under a normal distribution assumption. Using an ANOVA power approximation, the total sample size was estimated at approximately N = 30, or n = 10 participants per group. However, to account for the non-parametric nature of the Kruskal–Wallis test, Monte Carlo simulations with 5000 iterations were performed, assuming equally spaced group means (0, −d/2, −d) with a standard deviation of 1.0. These simulations confirmed that a sample size of 10 per group yields a power of approximately 0.75, falling short of the target, whereas 11 per group (total N = 33) achieves a power of approximately 0.81, meeting or exceeding the desired threshold. Thus, a minimum of 11 participants per group is recommended to ensure sufficient statistical power for detecting the anticipated large effects.

#### 4.4.2. Statistical Tool

Analyses were conducted using the R Statistical language (version 4.3.3; R Core Team, Vienna, Austria [89]) on Windows 11 Pro 64 bit (build 26100), using the packages ggpmisc (version 0.6.1; [90]), ggpp (version 0.5.8.1; [91]), ggpubr (version 0.6.0; [92]), report (version 0.5.8; [93]), correlation (version 0.8.5; [94]), gtsummary (version 1.7.2; [95]), ggplot2 (version 3.5.0; [96]), dplyr (version 1.1.4; [97]) and tidyr (version 1.3.1; [88,98]).

## 5. Conclusions

The study results confirm that PTSD is associated with a characteristic profile of neurobiological dysregulation and profound deficits in existential attitudes. In individuals with PTSD of ≤5 years’ duration, decreased levels of serotonin, noradrenaline, and cortisol were observed, accompanied by an increase in IL-12, suggesting the presence of an acute neuroinflammatory response and disturbances of the hypothalamic–pituitary–adrenal (HPA) axis. Participants with PTSD lasting >5 years showed significantly elevated cortisol levels and persistent psychological deficits, particularly in life control, purposefulness, and cognitive coherence. Clinically significant correlations were found between stress biomarkers and LAP-R scores—especially cortisol with Life Control and Death Acceptance, and noradrenaline with the Balance of Life Attitudes—indicating that biological parameters reflect not only the physiological state but also the patient’s psychological and identity structure. The variation in these relationships depending on PTSD duration and patient age highlights the dynamic nature of changes and the need for individualized therapeutic approaches. From a clinical perspective, the findings indicate that biomarkers (cortisol, noradrenaline, IL-12) may be used as diagnostic and prognostic indicators in PTSD as well as potential targets for intervention. At the same time, LAP-R scales may serve to monitor treatment progress in cognitive–existential integration. The results justify the implementation of complex therapies combining pharmacological measures (e.g., modulation of the HPA axis) with interventions aimed at restoring meaning in life, a sense of agency, and adaptation to the awareness of mortality.

## Figures and Tables

**Figure 1 ijms-26-09636-f001:**
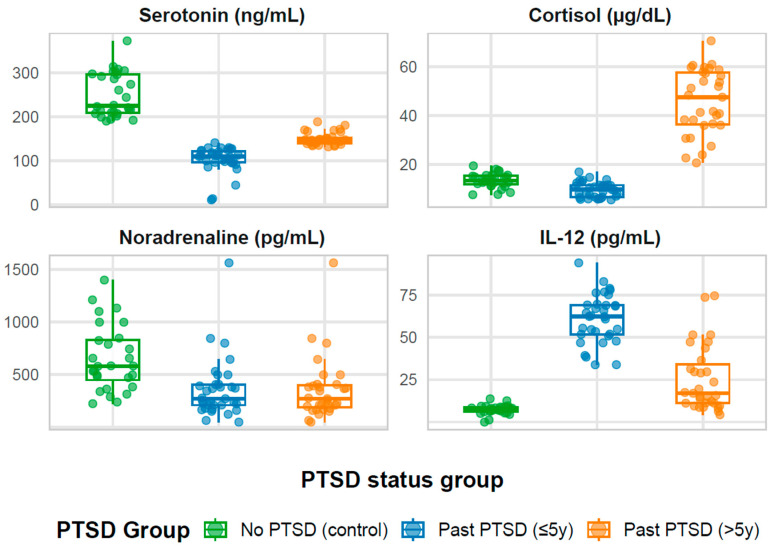
Distribution of biomarkers across PTSD status groups (boxplots with jittered points showing individual data distributions).

**Figure 2 ijms-26-09636-f002:**
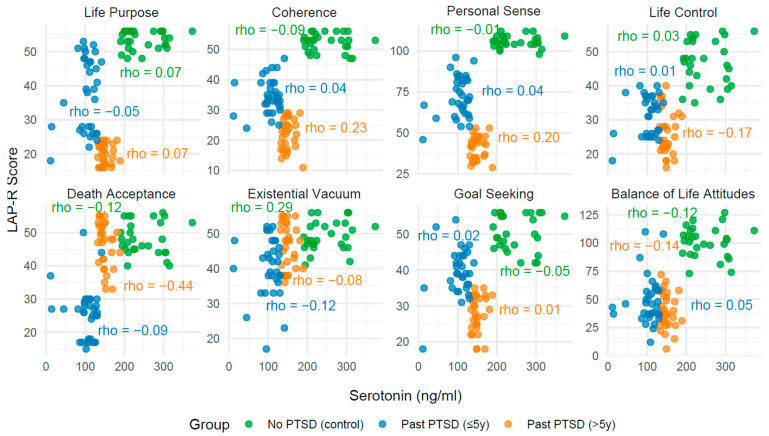
Scatterplots between serotonin levels and LAP-R scales across PTSD status.

**Figure 3 ijms-26-09636-f003:**
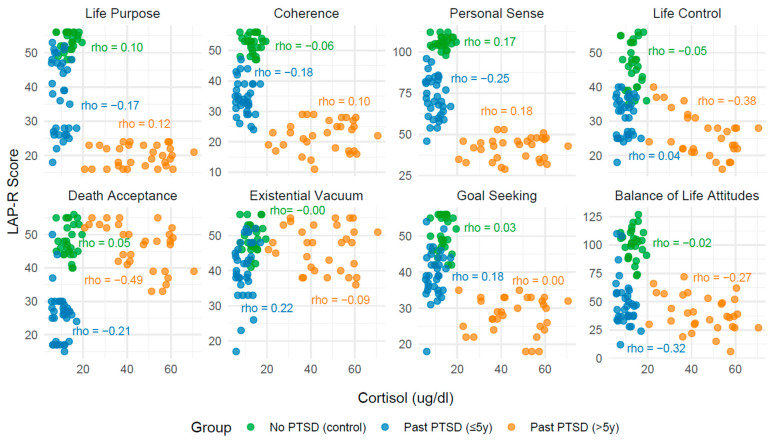
Scatterplots between cortisol levels and LAP-R scales across PTSD status.

**Figure 4 ijms-26-09636-f004:**
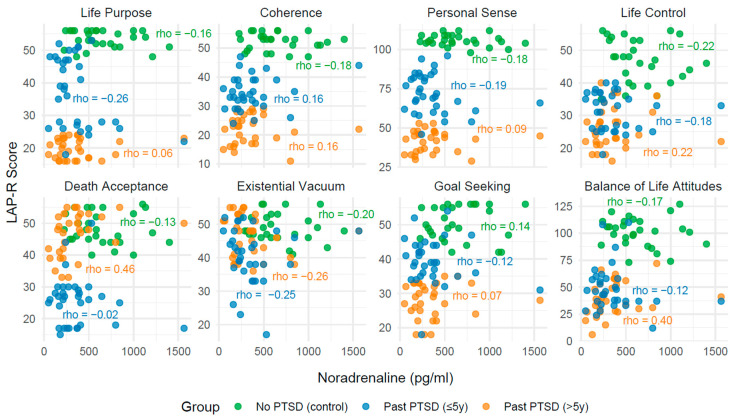
Scatterplots between noradrenaline levels and LAP-R scales across PTSD status.

**Figure 5 ijms-26-09636-f005:**
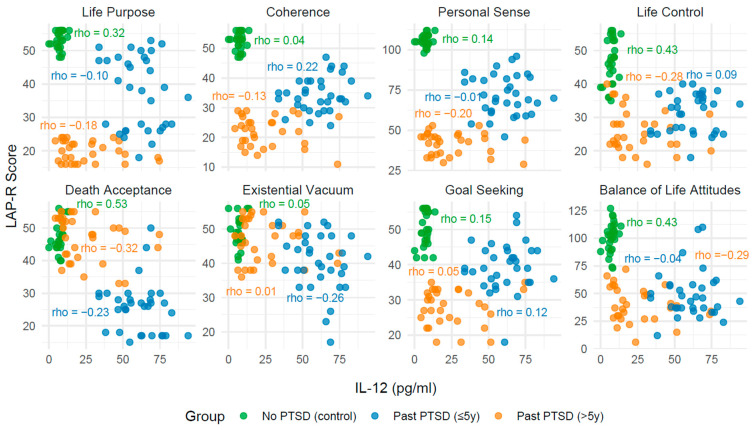
Scatterplots between IL-12 levels and LAP-R scales across PTSD status.

**Table 1 ijms-26-09636-t001:** Demographic, lifestyle, biomarker, and psychological characteristics of male mine rescue workers and former miners by PTSD Status (N = 92).

Characteristic	Overall (N = 92)	No PTSD (Control) (N = 28)	Past PTSD (≤5y) (N = 33)	Past PTSD (>5y) (N = 31)	*p*-Value
Demographics					
Age, median (IQR)	34.0 (28.8, 41.0)	33.5 (24.2, 41.5)	34.0 (31.0, 41.0)	36.0 (29.5, 41.0)	0.524
Biomarkers, median (IQR)					
Serotonin (ng/mL)	144.8 (120.2, 203.4)	225.2 ^a^ (209.6, 296.7)	109.9 ^c^ (96.6, 122.2)	147.4 ^b^ (139.6, 153.2)	<0.001
Cortisol (µg/dL)	14.0 (10.9, 36.2)	13.5 ^b^ (12.0, 15.4)	9.8 ^b^ (6.8, 11.6)	47.5 ^a^ (36.4, 57.6)	<0.001
Noradrenaline (pg/mL)	371.3 (220.3, 544.8)	580.2 ^a^ (449.1, 830.2)	271.7 ^b^ (210.7, 406.3)	271.7 ^a^ (189.0, 399.7)	<0.001
IL-12 (pg/mL)	21.6 (8.5, 54.7)	7.7 ^c^ (6.3, 8.6)	62.4 ^a^ (51.8, 69.1)	17.0 ^b^ (11.2, 33.9)	<0.001
LAP-R questionnaire, median (IQR)					
Life Purpose (LF)	37.0 (22.0, 51.2)	54.0 ^a^ (51.8, 56.0)	39.0 ^b^ (27.0, 47.0)	20.0 ^c^ (17.0, 23.0)	<0.001
Coherence (C)	33.0 (25.0, 48.2)	53.0 ^a^ (50.8, 54.2)	34.0 ^b^ (31.0, 39.0)	23.0 ^c^ (17.5, 26.0)	<0.001
Personal Sense (PS)	69.5 (46.0, 103.2)	105.0 ^a^ (104.0, 109.0)	70.0 ^b^ (62.0, 82.0)	44.0 ^c^ (35.5, 46.5)	<0.001
Life Control (LC)	34.0 (25.0, 40.0)	47.5 ^a^ (41.5, 53.0)	33.0 ^b^ (26.0, 37.0)	24.0 ^c^ (22.0, 29.5)	<0.001
Death Acceptance (DA)	44.0 (28.0, 50.0)	47.5 ^a^ (44.8, 53.0)	26.0 ^c^ (18.0, 29.0)	48.0 ^a^ (41.5, 52.0)	<0.001
Existential Vacuum (EV)	47.0 (40.0, 51.0)	48.5 ^a^ (46.8, 52.2)	42.0 ^b^ (37.0, 48.0)	48.0 ^a^ (40.5, 52.0)	<0.001
Goal Seeking (GS)	39.0 (32.0, 47.0)	50.0 ^a^ (46.8, 55.2)	39.0 ^b^ (35.0, 44.0)	29.0 ^c^ (24.5, 32.5)	<0.001
Balance of Life Attitudes (BLA)	54.0 (37.0, 90.2)	102.5 ^b^ (90.8, 111.0)	46.0 ^b^ (37.0, 58.0)	37.0 ^b^ (29.0, 52.5)	<0.001

Notes: Continuous variables are reported as median (interquartile range, IQR); categorical variables are reported as n (%). Values are rounded to one decimal place for consistency. *p*-values were calculated using the Kruskal–Wallis test for continuous variables and chi-squared or Fisher’s exact test for categorical variables. A probability value (*p*-value) < 0.05 indicates statistical significance. Groups with different letters (a, b, c) differ significantly (*p* < 0.05) based on compact letter display (CLD) notation for post hoc tests following significant Kruskal–Wallis or chi-squared results. Groups sharing the same letter are not significantly different. Order of letters corresponds to Past PTSD (≤5y), Past PTSD (>5y), and No PTSD (Control), respectively. A dash (-) indicates no significant group differences or insufficient data for post hoc testing.

**Table 2 ijms-26-09636-t002:** Wilcoxon r effect sizes for pairwise comparisons of biomarkers and LAP-R scales across PTSD status groups.

Parameter	Control vs. ≤5y r	Control vs. >5y r	≤5y vs. >5y r
Biomarkers			
Serotonin	0.86	0.86	0.84
Cortisol	0.54	0.86	0.86
Noradrenaline	0.53	0.53	0.03
IL-12	0.86	0.69	0.72
LAP-R questionnaire			
Life Purpose	0.78	0.86	0.81
Coherence	0.85	0.86	0.79
Personal Sense	0.86	0.86	0.84
Life Control	0.79	0.83	0.41
Death Acceptance	0.81	0.10	0.80
Existential Vacuum	0.55	0.20	0.36
Goal Seeking	0.71	0.86	0.75
Balance of Life Attitudes	0.77	0.86	0.22

**Table 3 ijms-26-09636-t003:** Biomarker and psychological characteristics of male mine rescue workers and former miners by PTSD status and age group.

Characteristic	No PTSD (Control) (N = 28)	PTSD ≤ 5y (N = 33)	PTSD > 5y (N = 31)
Serotonin (ng/mL)			
18–35 yrs	219.1 (211.6, 266.8)	111.6 (95.3, 122.7)	143.8 (136.4, 158.8)
36–50 yrs	273.9 (208.3, 303.3)	104.6 (97.7, 120.8)	148.6 (142.8, 152.6)
*p*-value (adjusted)	1.000	0.829	0.374
Cortisol (ug/dL)			
18–35 yrs	12.7 (11.5, 14.2)	11.1 (7.8, 12.8)	36.1 (29.1, 39.2)
36–50 yrs	14.9 (12.7, 15.8)	8.2 (6.1, 11.3)	57.6 (53.2, 59.7)
*p*-value (adjusted)	0.294	0.378	0.003
Noradrenaline (pg/mL)			
18–35 yrs	427.2 (320.8, 582.8)	369.2 (220.3, 534.6)	371.3 (225.7, 571.2)
36–50 yrs	765.7 (544.8, 960.4)	239.7 (179.7, 381.0)	211.3 (176.3, 354.3)
*p*-value (adjusted)	0.021	0.480	0.165
IL-12 (pg/mL)			
18–35 yrs	8.4 (7.4, 9.2)	68.0 (51.9, 77.1)	15.7 (10.3, 31.6)
36–50 yrs	6.5 (5.4, 7.7)	55.4 (46.8, 65.9)	26.3 (11.9, 32.7)
*p*-value (adjusted)	0.033	0.174	1.000
LF (Life Purpose)			
18–35 yrs	56.0 (50.8, 56.0)	27.5 (26.0, 29.8)	18.0 (16.0, 22.5)
36–50 yrs	53.0 (52.0, 54.8)	47.0 (45.0, 50.0)	20.5 (18.8, 23.0)
*p*-value (adjusted)	0.678	< 0.001	0.648
C (Coherence)			
18–35 yrs	53.5 (50.5, 56.0)	35.5 (32.0, 39.0)	21.0 (18.0, 24.0)
36–50 yrs	53.0 (51.0, 53.0)	33.0 (29.0, 35.0)	24.5 (17.8, 27.3)
*p*-value (adjusted)	0.504	0.909	0.909
PS (Personal Sense)			
18–35 yrs	105.5 (104.0, 109.0)	63.0 (59.0, 67.5)	42.0 (34.0, 45.0)
36–50 yrs	105.0 (104.0, 106.8)	82.0 (73.0, 85.0)	46.0 (36.8, 48.0)
*p*-value (adjusted)	1.000	0.001	0.265
LC (Life Control)			
18–35 yrs	53.0 (48.0, 54.8)	30.0 (25.0, 34.3)	31.0 (22.0, 35.0)
36–50 yrs	42.5 (39.3, 46.0)	35.0 (27.0, 37.0)	23.0 (20.3, 28.0)
*p*-value (adjusted)	0.006	0.474	0.147
DA (Death Acceptance)			
18–35 yrs	49.0 (47.3, 53.0)	24.5 (17.0, 26.3)	52.0 (46.0, 53.0)
36–50 yrs	44.5 (44.0, 49.0)	28.0 (26.0, 30.0)	44.5 (38.5, 48.3)
*p*-value (adjusted)	0.111	0.027	0.021
EV (Existential Vacuum)			
18–35 yrs	48.0 (46.3, 52.8)	40.5 (36.8, 48.0)	46.0 (42.0, 49.5)
36–50 yrs	49.0 (47.0, 51.8)	42.0 (37.0, 45.0)	48.5 (39.5, 53.0)
*p*-value (adjusted)	1.000	1.000	1.000
GS (Goal Seeking)			
18–35 yrs	50.0 (47.3, 55.0)	43.0 (35.0, 44.5)	28.0 (26.0, 32.5)
36–50 yrs	50.5 (44.3, 56.0)	39.0 (37.0, 41.0)	30.0 (23.5, 32.3)
*p*-value (adjusted)	1.000	1.000	1.000
BLA (Balance of Life Attitudes)			
18–35 yrs	106.5 (102.0, 111.0)	37.0 (31.8, 38.0)	41.0 (32.0, 57.5)
36–50 yrs	93.5 (86.5, 102.8)	57.0 (48.0, 66.0)	36.5 (27.0, 48.3)
*p*-value (adjusted)	0.033	0.001	0.466

## Data Availability

All data and analysis are available within the manuscript or upon request to the corresponding author.

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
