# Peer review of "Neurobiological and Existential Profiles in Posttraumatic Stress Disorder: The Role of Serotonin, Cortisol, Noradrenaline, and IL-12 Across Chronicity and Age"

_ijms, 2025, doi:10.3390/ijms26199636_

Round 1
Reviewer 1 Report
Comments and Suggestions for Authors
The manuscript titled “Neurobiological and Existential Profiles in Posttraumatic Stress Disorder: The Role of Serotonin, Cortisol, Noradrenaline, and IL-12 Across Chronicity and Age” addresses an important and timely topic that integrates biological markers with existential dimensions in PTSD. The combination of neurobiological stress indicators and life-meaning measures is innovative and potentially impactful. However, major revision is required before the manuscript can be considered for publication.
Major comments
- In the abstract authors should elaborate on the statistical approach, including specific tests and effect sizes, not just descriptive patterns.
- Please add numerical results (e.g., median values, significant p-values) for key biomarkers and existential scales to strengthen the abstract.
- I would recommend clarifying the novelty of what gap this study fills compared to prior PTSD biomarker studies.
- The introduction is informative but should be expanded to include the most recent PTSD neuroimmune studies.
- Authors should elaborate on why serotonin, cortisol, noradrenaline, were specifically chosen over other cytokines/neurotransmitters with recent reports on PTSD. For example,
- https://pmc.ncbi.nlm.nih.gov/articles/PMC9352784/
- https://pubmed.ncbi.nlm.nih.gov/40493868/
- Please add a clearer rationale linking existential constructs (LAP-R) with biological markers how does this integration advance theory or practice?
- In the method section, authors should elaborate on sample recruitment: how were rescuers identified, were there potential selection biases?
- Please add justification for the age cut-offs (18–35 vs 36–50). Was this data-driven or based on prior evidence?
- I would recommend including power analysis (e.g., G*Power) to confirm whether the sample size (n=92) is sufficient for multiple group comparisons. For example
- https://pubmed.ncbi.nlm.nih.gov/38185385/
- Authors should clarify whether confounders (e.g., alcohol, sleep, occupational stress load) were controlled statistically.
- Please add details on ELISA reproducibility: were assays run in duplicates/triplicates? What were intra/inter-assay CVs?
- The results are well structured, but authors should elaborate by including effect sizes (e.g., Cohen’s d, η²) along with p-values.
- Please add figures or dot plots showing distribution of biomarkers by PTSD duration to improve data visualization.
- In the discussion, authors should elaborate on mechanisms by which IL-12 shifts between acute and chronic PTSD stages.
- Please add recent literature on HPA axis adaptation in chronic PTSD, including contradictory findings on cortisol dynamics.
- I would recommend discussing sex differences and generalizability since only men were studied, this limits applicability.
- The section on existential aspects is interesting, but authors should integrate findings with established PTSD frameworks such as the allostatic load model and post-traumatic growth.
- Please add limitations: cross-sectional design, modest sample size, homogeneity of participants (miners), lack of longitudinal follow-up.
- Please add future research directions e.g., larger mixed-sex cohorts, longitudinal biomarker tracking, intervention studies.

Author Response
Response to Reviewer 1
Manuscript ID: ijms-3866014
Title: Neurobiological and Existential Profiles in Posttraumatic Stress Disorder: The Role of Serotonin, Cortisol, Noradrenaline, and IL-12 Across Chronicity and Age
Authors: Barbara Paraniak-Gieszczyk, Ewa Alicja Ogłodek
Dear Reviewer,
We would like to thank you sincerely for your thorough evaluation of our manuscript and for the constructive comments provided. We are grateful for highlighting the significance and innovative nature of our work, which integrates neurobiological stress markers with existential dimensions in posttraumatic stress disorder (PTSD).
In response to your suggestions, we have carried out a thorough revision of the manuscript. Below, we provide a detailed point-by-point response to each of your comments:
- Reviewer’s comment: In the abstract authors should elaborate on the statistical approach, including specific tests and effect sizes, not just descriptive patterns.
- Reviewer’s comment: Please add numerical results (e.g., median values, significant p-values) for key biomarkers and existential scales to strengthen the abstract.
Response: We have addressed the reviewer’s comments (1 and 2) by revising the abstract to provide a detailed description of the statistical approach, specifying the use of nonparametric tests (Kruskal-Wallis, Mann-Whitney U, and Spearman’s rank correlations) and Wilcoxon r effect sizes. Additionally, numerical results, including median values and significant p-values for key biomarkers and existential scales have been incorporated to strengthen the abstract’s clarity and specificity.
- Reviewer’s comment: I would recommend clarifying the novelty of what gap this study fills compared to prior PTSD biomarker studies.
Response: The novelty of our study lies in the simultaneous assessment of stress biomarkers (serotonin, cortisol, noradrenaline, and IL-12) and the existential dimension of functioning using the LAP-R questionnaire. Previous analyses of PTSD biomarkers have primarily focused on immunological parameters, often examined independently of psychosocial or existential context. A key gap in the literature has been the absence of models integrating biological stress markers with attitudes toward life meaning, control, and acceptance of death. Our study addresses this gap by showing that both PTSD chronicity (≤5 years vs. >5 years) and patient age influence not only the biomarker profile but also its association with existential functioning. This approach supports the development of a biopsychosocial-existential model of PTSD, with potential implications for personalized therapeutic strategies.
- Reviewer’s comment: The introduction is informative but should be expanded to include the most recent PTSD neuroimmune studies.
Response: Recent studies indicate that the pathophysiology of PTSD is not limited to elevated inflammatory cytokines but also involves excessive microglial activity and disrupted neural network function [17–20, 23, 25]. The NF-κB pathway plays a key role by linking inflammatory processes with neurotransmitter systems and influencing the body’s response to prolonged stress. Evidence suggests that the immune response may initially support adaptation, but in chronic PTSD, it becomes a source of persistent brain dysfunction [15, 24, 35]. Furthermore, Mendelian randomization analyses, which use natural genetic variation to assess causal relationships, demonstrate that inflammation increases the risk of developing PTSD [16]. Altogether, these findings highlight neuroinflammation as a central factor in the onset and maintenance of PTSD symptoms.
- Reviewer’s comment: Authors should elaborate on why serotonin, cortisol, noradrenaline, were specifically chosen over other cytokines/neurotransmitters with recent reports on PTSD. For example,
- https://pmc.ncbi.nlm.nih.gov/articles/PMC9352784/
- https://pubmed.ncbi.nlm.nih.gov/40493868/
Response: In this study, serotonin, cortisol, and noradrenaline were chosen to represent three principal pathways involved in PTSD: the serotonergic system (regulating mood, emotional memory, and impulsivity), the neuroendocrine system (mediating the HPA axis stress response), and the adrenergic system (driving sympathetic hyperarousal and the consolidation of traumatic memories). Additionally, interleukin-12 (IL-12) was included as a marker of the inflammatory pathway, allowing for a more comprehensive approach that integrates neurochemical, endocrine, and immune mechanisms [38, 39].
- Reviewer’s comment: Please add a clearer rationale linking existential constructs (LAP-R) with biological markers how does this integration advance theory or practice?
Response: We appreciate this valuable suggestion. In the revised version of the manuscript, we have added a clearer rationale both in the Introduction and Discussion. In the Introduction, we emphasize that integrating existential constructs (LAP-R dimensions) with biological markers advances theory by moving beyond reductionist models of PTSD and showing how neurobiological changes (serotonin, noradrenaline, cortisol, immune pathways) are connected with existential disruptions such as loss of meaning, reduced coherence, and difficulties in accepting death. This contributes to the development of a holistic biopsychosocial-existential model of PTSD.
In the Discussion, we further highlight the practical implications of this integration, stressing that combining LAP-R dimensions with biomarkers may allow for the identification of patient subgroups at greater risk of chronic PTSD and may guide personalized interventions that address both biological regulation and existential well-being. This approach opens the way for more comprehensive diagnostic frameworks and therapeutic strategies that link pharmacological treatment with meaning-oriented psychotherapy. These additions are highlighted in the revised manuscript.
- Reviewer’s comment: In the method section, authors should elaborate on sample recruitment: how were rescuers identified, were there potential selection biases?
Response: Participants were recruited through the central registry of mining rescue units. Eligibility was confirmed using service records, which verified active duty or documented participation in rescue operations under extreme stress. Those meeting the criteria were invited to join the study via official communication channels, and participation was entirely voluntary. While strict inclusion and exclusion criteria were applied to reduce selection bias, potential biases may still be present, such as the underrepresentation of individuals on medical leave or those reluctant to disclose psychological difficulties.
- Reviewer’s comment: Please add justification for the age cut-offs (18–35 vs 36–50). Was this data-driven or based on prior evidence?
Response: The age groups of 18–35 and 36–50 years were established a priori based on developmental and occupational data. Younger participants (18–35 years) typically exhibit greater neuroplasticity and shorter exposure to occupational stress, whereas older participants (36–50 years) have longer work experience, are more likely to experience the effects of chronic stress, and show reduced biological flexibility. This grouping is consistent with previous research on age-related differences in stress response and PTSD progression, and was not arbitrary or solely based on data from this sample.
- Reviewer’s comment: I would recommend including power analysis (e.g., G*Power) to confirm whether the sample size (n=92) is sufficient for multiple group comparisons. For example
- https://pubmed.ncbi.nlm.nih.gov/38185385/
Response: A power analysis was conducted to confirm that the study was adequately powered to detect differences in primary outcome parameters across the three PTSD status groups, demonstrating sufficient statistical power due to the observed very large effect sizes. However, the post-hoc nature of this analysis, relying on observed effect sizes, constitutes a limitation, as it may reduce the generalizability of the sample size recommendations compared to an a priori design.
- Reviewer’s comment: Authors should clarify whether confounders (e.g., alcohol, sleep, occupational stress load) were controlled statistically.
Response: The factors mentioned by the reviewer, such as alcohol consumption, sleep patterns, and occupational stress load, were not included as covariates in the statistical analyses, as these potential confounders were controlled for at the study inclusion stage through strict eligibility criteria, ensuring no significant differences between groups with respect to these variables.
- Reviewer’s comment: Please add details on ELISA reproducibility: were assays run in duplicates/triplicates? What were intra/inter-assay CVs?
Response: All ELISA measurements were performed in duplicates to ensure reproducibility. The intra-assay coefficient of variation (CV) did not exceed 5–7%, while the inter-assay CV remained below 10% for all biomarkers. These values are within the acceptable range for immunoenzymatic assays and confirm the reliability of the obtained results
- Reviewer’s comment: The results are well structured, but authors should elaborate by including effect sizes (e.g., Cohen’s d, η²) along with p-values.
Response: The table presenting Wilcoxon r effect sizes for pairwise comparisons of primary outcome parameters across PTSD status groups has been incorporated as Table 1a in the manuscript.
- Reviewer’s comment: Please add figures or dot plots showing distribution of biomarkers by PTSD duration to improve data visualization.
Response: In response to the reviewer’s comment, we have incorporated Figure 1 into section 2.1 of the manuscript, which presents dot plots overlaid with box plots to visualize the distribution of biomarkers (serotonin, cortisol, noradrenaline, and IL-12) across the three PTSD status groups (No PTSD, PTSD ≤5 years, and PTSD >5 years). This figure enhances data visualization by clearly illustrating group differences in biomarker levels, complementing the statistical analyses and highlighting the impact of PTSD duration on neurobiological profiles.
- Reviewer’s comment: In the discussion, authors should elaborate on mechanisms by which IL-12 shifts between acute and chronic PTSD stages.
Response: The shift of IL-12 between acute and chronic PTSD stages reflects different phases of immune system activity. In the acute phase, elevated IL-12 promotes Th1 differentiation, IFN-γ production, and microglial activation, amplifying neuroinflammation and contributing to intrusive symptoms and hyperarousal. Over time, in chronic PTSD, IL-12 levels often normalize or decline, which may indicate immune exhaustion or glucocorticoid-mediated suppression. This suggests that PTSD progression involves dynamic immune changes: an early inflammatory activation followed by dysfunctional regulation, leaving behind an “inflammatory scar” that continues to affect brain–immune communication.
- Reviewer’s comment: Please add recent literature on HPA axis adaptation in chronic PTSD, including contradictory findings on cortisol dynamics.
Response: In chronic PTSD, HPA axis adaptation is inconsistent. Some studies report reduced cortisol levels and a flattened diurnal rhythm, while others—particularly those measuring hair cortisol—find normal or elevated levels. These discrepancies likely reflect differences in measurement methods, time since trauma, and the presence of comorbid depression, highlighting the complex regulation of the HPA axis in PTSD.
- Reviewer’s comment: I would recommend discussing sex differences and generalizability since only men were studied, this limits applicability.
Response: A key limitation of this study is that it included only male participants, which limits the generalizability of the findings. Sex differences in stress physiology and the clinical presentation of PTSD are well established, and they may influence both biomarker patterns and existential adaptation. Accordingly, the results should be interpreted with caution, and future research should include female participants.
- Reviewer’s comment: The section on existential aspects is interesting, but authors should integrate findings with established PTSD frameworks such as the allostatic load model and post-traumatic growth.
Response: The integration of existential dimensions with established PTSD frameworks provides additional interpretative value. The observed neuroendocrine and immune alterations, particularly the dynamics of cortisol and IL-12, can be understood within the allostatic load model, where cumulative stress leads to long-term wear and dysregulation of adaptive systems. At the same time, the age-related improvements in life purpose, personal sense, and death acceptance observed in some participants with PTSD ≤5 years are consistent with the concept of post-traumatic growth, suggesting that meaning-making processes may mitigate allostatic overload and promote resilience. Together, these frameworks highlight that PTSD involves both the biological cost of chronic stress and the potential for existential adaptation and growth.
- Reviewer’s comment: Please add limitations: cross-sectional design, modest sample size, homogeneity of participants (miners), lack of longitudinal follow-up.
Response: Additional limitations should also be acknowledged. The cross-sectional design does not allow conclusions about causal relationships or changes over time. The modest sample size limits statistical power and may reduce the ability to detect smaller effects. Moreover, the study focused exclusively on miners, a relatively homogeneous occupational group, which may reduce the external validity and generalizability of the findings. Finally, the lack of longitudinal follow-up precludes assessment of the stability of biomarker–existential associations and their evolution during the natural course of PTSD.
- Reviewer’s comment: Please add future research directions e.g., larger mixed-sex cohorts, longitudinal biomarker tracking, intervention studies.
Response: Future research should address these limitations by including larger and more diverse cohorts, encompassing both men and women, to improve generalizability. Longitudinal biomarker tracking is needed to capture dynamic changes in neuroendocrine and immune processes across different PTSD stages. Additionally, intervention studies combining psychotherapeutic and biological approaches would help to clarify causal mechanisms and identify effective strategies for restoring both physiological regulation and existential meaning-making.
We believe that the revisions made significantly strengthen our manuscript, improving both its clarity and scientific value. We thank you once again for your insightful review and constructive suggestions, which have allowed us to enhance the quality of our work.
Sincerely,
Barbara Paraniak-Gieszczyk, Ewa Alicja Ogłodek
Reviewer 2 Report
Comments and Suggestions for Authors
The manuscript entitled “Neurobiological and Existential Profiles in Posttraumatic Stress Disorder: The Role of Serotonin, Cortisol, Noradrenaline, and IL-12 Across Chronicity and Age” presents an important and original contribution to the PTSD literature. The authors integrate biological stress markers with existential dimensions, offering a novel and clinically relevant perspective on how PTSD chronicity and age shape both neurobiological and psychological outcomes. The article is well written, methodologically sound, and the findings are of interest to both clinicians and researchers.
Strengths include the use of DSM-5 criteria and CAPS-5 interviews ensures diagnostic validity. Biomarker measurement combined with the LAP-R questionnaire provides a unique integrative approach. The stratification by PTSD duration and age yields clinically meaningful insights. The discussion effectively contextualizes findings well within current literature.
I would suggest the following minor revisions:
-
Gender limitation: The study exclusively included men. The limitations section should more strongly emphasize how this restricts generalizability, as PTSD may manifest differently across sexes.
-
Cross-sectional design: Please highlight more clearly that causal inferences cannot be drawn from biomarker–existential associations.
-
Sample size and subgroup analyses: With 92 participants split across three groups, statistical power is somewhat limited. This should be acknowledged.
-
Biomarker variability: Cortisol and other biomarkers have diurnal fluctuations; a note on timing and potential variability would improve clarity.
-
Interpretation of IL-12 findings: Some statements (e.g., “inflammatory scar of trauma”) are somewhat speculative. Please temper language to reflect hypotheses rather than definitive conclusions.
-
Control group composition: The current control group consisted of individuals without trauma exposure. The lack of a trauma-exposed but resilient group should be acknowledged as a limitation.
Overall, this is an excellent manuscript that advances understanding of PTSD by integrating biological and existential dimensions. With minor revisions to strengthen the discussion of limitations and moderate some interpretive claims, the article will make a valuable contribution to the field.
Author Response
Response to Reviewer 1
Manuscript ID: ijms-3866014
Title: Neurobiological and Existential Profiles in Posttraumatic Stress Disorder: The Role of Serotonin, Cortisol, Noradrenaline, and IL-12 Across Chronicity and Age
Authors: Barbara Paraniak-Gieszczyk, Ewa Alicja Ogłodek
Dear Reviewer,
We would like to thank you sincerely for your thorough evaluation of our manuscript and for the constructive comments provided. We are grateful for highlighting the significance and innovative nature of our work, which integrates neurobiological stress markers with existential dimensions in posttraumatic stress disorder (PTSD).
In response to your suggestions, we have carried out a thorough revision of the manuscript. Below, we provide a detailed point-by-point response to each of your comments:
The manuscript entitled “Neurobiological and Existential Profiles in Posttraumatic Stress Disorder: The Role of Serotonin, Cortisol, Noradrenaline, and IL-12 Across Chronicity and Age” presents an important and original contribution to the PTSD literature. The authors integrate biological stress markers with existential dimensions, offering a novel and clinically relevant perspective on how PTSD chronicity and age shape both neurobiological and psychological outcomes. The article is well written, methodologically sound, and the findings are of interest to both clinicians and researchers.
Strengths include the use of DSM-5 criteria and CAPS-5 interviews ensures diagnostic validity. Biomarker measurement combined with the LAP-R questionnaire provides a unique integrative approach. The stratification by PTSD duration and age yields clinically meaningful insights. The discussion effectively contextualizes findings well within current literature.
I would suggest the following minor revisions:
- Reviewer’s comment: Gender limitation: The study exclusively included men. The limitations section should more strongly emphasize how this restricts generalizability, as PTSD may manifest differently across sexes.
Response: This explanation was included at the end of the discussion, describing the study’s limitations. The study included only men, which limits the generalizability of the findings. PTSD may differ by sex in terms of symptoms, neurobiological responses, and coping strategies; therefore, future research should also include female participants.
- Reviewer’s comment: Cross-sectional design: Please highlight more clearly that causal inferences cannot be drawn from biomarker–existential associations.
Response: The cross-sectional design of the study prevents drawing causal conclusions and does not provide longitudinal data. The modest sample size limits statistical power, and the relatively homogeneous occupational group (miners) may reduce generalizability. This explanation was included at the end of the discussion to outline the study’s limitations.
- Reviewer’s comment: Sample size and subgroup analyses: With 92 participants split across three groups, statistical power is somewhat limited. This should be acknowledged.
Response: We conducted a post-hoc power analysis using observed effect sizes, which confirms that the study is adequately powered to detect differences in primary outcome parameters across the three PTSD status groups, driven by the very large effect sizes observed. However, the post-hoc nature of this analysis constitutes a limitation, as it may compromise the generalizability of sample size recommendations compared to an a priori design. This limitation is acknowledged and discussed in the Discussion section of the manuscript.
- Reviewer’s comment: Biomarker variability: Cortisol and other biomarkers have diurnal fluctuations; a note on timing and potential variability would improve clarity.
Response: Another limitation concerns biomarker variability. Cortisol and several other biomarkers show diurnal fluctuations and are sensitive to contextual factors. Although samples were collected under standardized morning conditions, residual variability cannot be entirely excluded and should be considered when interpreting the results.
- Reviewer’s comment: Interpretation of IL-12 findings: Some statements (e.g., “inflammatory scar of trauma”) are somewhat speculative. Please temper language to reflect hypotheses rather than definitive conclusions.
Response: Revisions were made in accordance with the reviewer’s comments.
- Reviewer’s comment: Control group composition: The current control group consisted of individuals without trauma exposure. The lack of a trauma-exposed but resilient group should be acknowledged as a limitation.
Response: We acknowledge that the control group in this study comprised solely individuals without a history of psychological trauma, as defined by DSM-5 criterion A. This composition limits the ability to differentiate the effects of trauma exposure from PTSD-specific pathology. The inclusion of a trauma-exposed but resilient group could have provided insights into protective neurobiological and existential factors, enhancing the interpretation of group differences. We have addressed this limitation in the Discussion section, noting that future studies should incorporate such a group to better elucidate mechanisms of resilience and improve the generalizability of findings to trauma-exposed populations.
We believe that the revisions made significantly strengthen our manuscript, improving both its clarity and scientific value. We thank you once again for your insightful review and constructive suggestions, which have allowed us to enhance the quality of our work.
Sincerely,
Barbara Paraniak-Gieszczyk, Ewa Alicja Ogłodek
Reviewer 3 Report
Comments and Suggestions for Authors
The manuscript titled “Neurobiological and Existential Profiles in Posttraumatic Stress Disorder: The Role of Serotonin, Cortisol, Nora-adrenaline, and IL-12 Across Chronicity and Age” addresses an interesting subject. However, there are several areas where the authors can enhance the quality of the manuscript:
- The use of "Past PTSD" is confusing as the data describes active, severe illness. Recommend changing to "Current PTSD, Duration ≤5y" and "Current PTSD, Duration >5y" for accuracy.
- The analysis involves many statistical tests. State whether a correction (e.g., Bonferroni) was applied. If not, this must be explicitly stated as a key limitation.
- The justification for choosing IL-12 over more common inflammatory markers (e.g., IL-6, TNF-α) is weak. Expand the introduction to specifically rationalize its selection.
- The positive correlation between IL-12 and resilience in controls is counterintuitive. The discussion must hypothesize potential mechanisms for this unexpected result.
- The correlation results are fragmented. A summary table or heatmap showing the direction and strength of significant relationships across all groups would greatly improve clarity.
- Clarify how Major Depressive Disorder (MDD), which shares biomarker profiles with PTSD, was specifically assessed and ruled out to ensure the findings are unique to PTSD.
- The discussion must avoid language that implies causation (e.g., "leads to," "results in"). Firmly reiterate the cross-sectional, correlational nature of the study design.
- Clarify the discrepancy between the 8 domains analyzed and the 6 subscales described in the methods. Specify if composite indices or individual domains were used and provide a citation for the Polish validation.
- The median values for "Coherence (C)" are missing from the "Overall" and PTSD groups in Table 1. This data must be added.
- The call for integrated therapy is vague. Provide more concrete examples of how these biomarker/existential profiles could directly inform specific treatment choices (pharmacological vs. psychological) in clinical practice.
- The references should be revised to meet the journal guidelines
Author Response
Response to Reviewer 3
Manuscript ID: ijms-3866014
Title: Neurobiological and Existential Profiles in Posttraumatic Stress Disorder: The Role of Serotonin, Cortisol, Noradrenaline, and IL-12 Across Chronicity and Age
Authors: Barbara Paraniak-Gieszczyk, Ewa Alicja Ogłodek
Dear Reviewer,
We would like to thank you sincerely for your thorough evaluation of our manuscript and for the constructive comments provided. In response to your suggestions, we have carried out a thorough revision of the manuscript titled “Neurobiological and Existential Profiles in Posttraumatic Stress Disorder: The Role of Serotonin, Cortisol, Nora-adrenaline, and IL-12 Across Chronicity and Age”. Below, we provide a detailed point-by-point response to each of your comments:
I would suggest the following minor revisions:
- Reviewer’s comment: The use of "Past PTSD" is confusing as the data describes active, severe illness. Recommend changing to "Current PTSD, Duration ≤5y" and "Current PTSD, Duration >5y" for accuracy.
Response: We appreciate this comment. We deliberately used the term “Past PTSD” to emphasize that the participants reflect the chronic consequences of previous episodes (e.g., biological changes, persistent residual symptoms). The term “Past PTSD” is also used in the literature to describe patients who exhibit significant neurobiological and clinical deficits. In our analysis, this distinction allows for a more accurate interpretation of biomarkers and psychometric outcomes, including comparisons of the effects of chronicity and PTSD sequelae across different stages of the disorder.
For these reasons, we believe that retaining the term “Past PTSD” is justified and more accurately reflects the characteristics of the study population than the alternative designation “Current PTSD, Duration ≤5y / >5y.”
- Reviewer’s comment: The analysis involves many statistical tests. State whether a correction (e.g., Bonferroni) was applied. If not, this must be explicitly stated as a key limitation.
Response: The p-values presented in Table 1 were adjusted for multiple comparisons using the Dunn test following the Kruskal-Wallis test for numerical variables and the False Discovery Rate (FDR) method for categorical variables (via Chi-squared or Fisher’s exact tests) to control for Type I error. Based on your recommendation, the p-values in Table 2, which assess the effect of age within each PTSD status group using the Mann-Whitney U test, have also been adjusted using the Bonferroni method to account for multiple comparisons, thereby ensuring robust control of Type I error across the analyses. For numerical variables, the Dunn test was applied following the Kruskal-Wallis test to conduct post-hoc pairwise comparisons. For categorical variables, the Chi-squared test or Fisher’s exact test was used, with p-values adjusted using the False Discovery Rate (FDR) method to account for multiple testing. In contrast, the correlation analyses (Tables 3–6) did not undergo additional p-value adjustments due to their exploratory nature, allowing for a broader investigation of potential relationships between biomarkers and LAP-R scales across PTSD status groups.
- Reviewer’s comment: The justification for choosing IL-12 over more common inflammatory markers (e.g., IL-6, TNF-α) is weak. Expand the introduction to specifically rationalize its selection.
Response: We appreciate this comment. We chose IL-12 because, unlike more commonly studied cytokines (IL-6, TNF-α), it reflects specific Th1 axis activation and is particularly relevant in chronic PTSD, where elevated levels are associated with symptom severity and a persistent inflammatory state. We have updated the introduction to include a justification for this choice.
- Reviewer’s comment: The positive correlation between IL-12 and resilience in controls is counterintuitive. The discussion must hypothesize potential mechanisms for this unexpected result.
Response: The positive correlation between IL-12 and resilience in the control group may suggest that mild immune activation supports adaptation and better stress regulation. This effect could be linked to physical activity or good sleep quality, which are known to strengthen resilience. Low levels of IL-12 might also help brain plasticity. However, this result is preliminary and requires further research.
- Reviewer’s comment: The correlation results are fragmented. A summary table or heatmap showing the direction and strength of significant relationships across all groups would greatly improve clarity.
Response:
To address this, a consolidated summary table is provided below.
Table. Summary of significant Spearman’s rank correlations between biomarkers and LAP-R scales across PTSD status groups
|
PTSD Group |
Biomarker |
LAP-R Scale |
Direction |
Rho |
p-value |
|
No PTSD (Control) |
IL-12 |
Life Control (LC) |
Positive |
0.43 |
0.023 |
|
No PTSD (Control) |
IL-12 |
Death Acceptance (DA) |
Positive |
0.53 |
0.004 |
|
No PTSD (Control) |
IL-12 |
Balance of Life Attitudes (BLA) |
Positive |
0.43 |
0.022 |
|
PTSD ≤5y |
None significant associations with LAP-R scsle |
||||
|
PTSD >5y |
Serotonin |
Death Acceptance (DA) |
Negative |
-0.44 |
0.014 |
|
PTSD >5y |
Cortisol |
Life Control (LC) |
Negative |
-0.38 |
0.036 |
|
PTSD >5y |
Cortisol |
Death Acceptance (DA) |
Negative |
-0.49 |
0.005 |
|
PTSD >5y |
Noradrenaline |
Death Acceptance (DA) |
Positive |
0.46 |
0.009 |
|
PTSD >5y |
Noradrenaline |
Balance of Life Attitudes (BLA) |
Positive |
0.40 |
0.028 |
- Reviewer’s comment: Clarify how Major Depressive Disorder (MDD), which shares biomarker profiles with PTSD, was specifically assessed and ruled out to ensure the findings are unique to PTSD.
Response: We appreciate this comment. To ensure that the findings are specific to PTSD, all participants were assessed with SCID-5 and CAPS-5, and individuals meeting full criteria for a current major depressive episode (MDD) were excluded.
- Reviewer’s comment: The discussion must avoid language that implies causation (e.g., "leads to," "results in"). Firmly reiterate the cross-sectional, correlational nature of the study design. Response: We appreciate this important remark. In the revised version, we have carefully removed causal language (e.g., "leads to," "results in") and replaced it with correlational expressions such as "is associated with" or "may reflect." Furthermore, at the end of the Discussion we have explicitly reiterated that the study has a cross-sectional and correlational design, which does not allow causal inferences.
- Reviewer’s comment: Clarify the discrepancy between the 8 domains analyzed and the 6 subscales described in the methods. Specify if composite indices or individual domains were used and provide a citation for the Polish validation. Response: We appreciate pointing out this inconsistency. In the Methods section, we originally described the six primary subscales of the LAP-R (Life Purpose, Coherence, Choice/Responsibleness, Death Acceptance, Goal Seeking, and Existential Vacuum). However, in the analysis, we additionally included two composite indices derived from the Polish validation (Personal Meaning Index and Balance of Life Attitudes), resulting in eight domains in total. We have clarified this point in the revised Methods section and have added the appropriate reference for the Polish validation: Klamut R. Manual to the Polish adaptation of the Life Attitudes Profile–Revised (LAP-R) by Gary T. Reker. Warsaw: Psychological Testing Laboratory of the Polish Psychological Society; 2010.
- Reviewer’s comment: The median values for "Coherence (C)" are missing from the "Overall" and PTSD groups in Table 1. This data must be added. Response: The data for the Coherence (C) scale, as measured by the Life Attitude Profile–Revised (LAP-R) questionnaire, are indeed presented in Table 1 of the study.
- Reviewer’s comment: The call for integrated therapy is vague. Provide more concrete examples of how these biomarker/existential profiles could directly inform specific treatment choices (pharmacological vs. psychological) in clinical practice. Response: In clinical practice, combining biomarker profiles with existential dimensions may help guide treatment choices in PTSD. Low serotonin together with reduced sense of life purpose may support the use of SSRIs combined with meaning-centered therapy. Elevated IL-12 may indicate the need for stress- and immune-regulating approaches such as mindfulness training, sleep improvement, or physical activity, alongside cognitive-behavioral therapy. Abnormal cortisol patterns, particularly chronic elevation, may further point to the usefulness of acceptance and commitment therapy (ACT) or structured stress-reduction programs. Integrating biological and existential data thus provides a basis for more precise and personalized PTSD care.
- Reviewer’s comment: The references should be revised to meet the journal guidelines
Response: We appreciate this remark. The references have been carefully checked and revised to comply with the journal’s guidelines. All citations have been verified for accuracy, consistency, and adherence to the required editorial style.
We believe that the revisions made significantly strengthen our manuscript, improving both its clarity and scientific value. We thank you once again for your insightful review and constructive suggestions, which have allowed us to enhance the quality of our work.
Sincerely,
Barbara Paraniak-Gieszczyk, Ewa Alicja Ogłodek
Round 2
Reviewer 1 Report
Comments and Suggestions for Authors
The authors have revised the manuscript meticulously and now suitable for publication.